# Impact of reduced anthropogenic emissions during COVID-19 on air quality in India

Mengyuan Zhang[1], Arpit Katiyar[2], Shengqiang Zhu[1], Juanyong Shen[3], Men Xia[4], Jinlong Ma[1], Sri Harsha Kota[2], Peng Wang[4], Hongliang Zhang[1,5]

[1]Department of Environmental Science and Engineering, Fudan University, Shanghai 200438, China
[2]Department of Civil Engineering, Indian Institute of Technology Delhi, 110016, India
[3]School of Environmental Science and Engineering, Shanghai Jiao Tong University, Shanghai 200240, China
[4]Department of Civil and Environmental Engineering, The Hong Kong Polytechnic University, Hong Kong SAR, 99907, China
[5]Institute of Eco-Chongming (IEC), Shanghai 200062, China

*Correspondence to*: Peng Wang (peng.ce.wang@polyu.edu.hk); Hongliang Zhang (zhanghl@fudan.edu.cn)

**Abstract.** To mitigate the impacts of the pandemic of coronavirus disease 2019 (COVID-19), the Indian government implemented lockdown measures on March 24, 2020, which prohibit unnecessary anthropogenic activities and thus leading to a significant reduction in emissions. To investigate the impacts of this lockdown measure on air quality in India, we used the Community Multi-Scale Air Quality (CMAQ) model to estimate the changes of key air pollutants. From pre-lockdown to lockdown periods, improved air quality is observed in India, indicated by the lower key pollutant levels such as $PM_{2.5}$ (-26%), maximum daily 8-h average ozone (MDA8 $O_3$) (-11%), $NO_2$ (-50%), and $SO_2$ (-14%). In addition, changes in these pollutants show distinct spatial variations with the more important decrease in northern and western India. During the lockdown, our results illustrate that such emission reductions play a positive role in the improvement of air quality. Significant reductions of $PM_{2.5}$ concentration and its major components are predicted, especially for secondary inorganic aerosols that are up to 92%, 57%, and 79% for nitrate ($NO_3^-$), sulfate ($SO_4^{2-}$), ammonium ($NH_4^+$), respectively. On average, the MDA8 $O_3$ also decreases 15% during the lockdown period although it increases sparsely in some VOC-limited urban locations, which is mainly due to the more significant reduction of $NO_x$ than VOCs. More aggressive and localized emissions control strategies should be implemented in India to mitigate air pollutions in the future.

## 1 Introduction

India, the second-most populous country in the world, has been suffered from severe air pollution along with rapid urbanization and industrialization in recent decades (Karambelas et al., 2018), and 13 Indian cities were among the world's top 20 most polluted cities according to the World Health Organization (WHO) (WHO, 2018). High-level pollution leads to health risks and ecosystem damages, which caused 1.24 million deaths in India in 2017 (Balakrishnan et al., 2019) and a great loss of crops (Oksanen et al., 2013; Lal et al., 2017). To mitigate air pollution, the Indian government has been promoting effective emission control strategies such as the conversion of fossil fuels to clean fuels in the nationwide campaign Clean India Mission (CIM).

However, such long-term or short-term reduction strategies seem to show insufficiency in the restoration of ambient air quality (Beig et al., 2013; Purohit et al., 2019; Banerjee et al., 2017).

Due to the pandemic of coronavirus disease 2019 (COVID-19), nationwide or partial lockdown measures have been implemented in many countries (Chintalapudi et al., 2020; Dantas et al., 2020; Ehrlich et al., 2020). Indian government declared corresponding bans since the detection of the first confirmed case on January 30, 2020. Then, to counter the fast contagion of COVID-19, a 3-week nationwide lockdown was imposed in India on March 24, which was expended till June 30. The lockdown measures mitigate the impact of COVID-19 on Indian health infrastructure and it also helped in curbing the rate of the spread of this infectious disease among people (Pai et al., 2020; Anderson et al., 2020). Because of the prohibition of industrial activities and mass transportation, anthropogenic emissions showed a tremendous reduction. Besides, several studies showed that dramatic emission reductions had an enormous impact on the formation of air pollution and positively influence air quality (Isaifan, 2020; Bao and Zhang, 2020; Gautam, 2020). Thus, the lockdown also provides a valuable opportunity to assess the changes in air pollutants with significantly reduced anthropogenic emissions in a short time.

Conspicuous reductions in concentrations of pollutants were also claimed in different regions (Otmani et al., 2020; Dantas et al., 2020; Nakada and Urban, 2020). Most Indian studies claimed the greatest reduction of particulate matter with an aerodynamic diameter of less than 2.5 μm ($PM_{2.5}$), up to 50% (Kumar et al., 2020; Mahato et al., 2020; Sharma et al., 2020). However, an increase in ozone ($O_3$) concentrations was observed (Collivignarelli et al., 2020; Sicard et al., 2020) and severe air pollution events still occurred after large emissions reduction due to unfavorable meteorological conditions (Wang et al., 2020). Moreover, another analysis showed that the effects of lockdown during the COVID-19 pandemic on $PM_{2.5}$ and $O_3$ pollution levels were less than the expected response to the enacted stay-at-home order (Bujin et al., 2020). Hence, the significance and impacts of lockdown measures are still not well understood.

Therefore, it is significant to understand the mechanisms involving in air pollution formation before and after dramatic emission changes comprehensively, in addition to the comparison of air pollution levels. Mahato et al. (2020) concluded that air quality in India from March 24 to April 14 was improved sharply according to the change of the National Air Quality Index, especially for Delhi. Srivastava et al. (2020) reported the concentrations of primary air pollutants are drastically lowed as a result of emission reduction. Kumari and Toshniwal (2020) also stated that the concentration of key pollutants such as $PM_{2.5}$ in both Delhi and Mumbai shows a decreasing trend. These studies pointed out that the air quality was improved during the lockdown period compared with the period before lockdown and depends on the duration of the lockdown (Kumar et al., 2020; Mor et al., 2021). Besides, compared with the same period in previous years, Gautam (2020) claimed that aerosol concentration levels are at their lowest in the last 20 years during lockdown based on satellite data. Selvam et al. (2020) stated that Air Quality Index (AQI) was improved by 58% in Gujarat state of western India during lockdown (March 24, 2020 – April 20, 2020) compared to 2019. Kabiraj and Gavli (2020) concluded that the mean concentration of $PM_{2.5}$ decreased by 42.25% from January to May in 2020 compared with 2019. Similarly, Das et al. (2020) also showed that great reductions of $PM_{2.5}$ were found across cities in the Indo-Gangetic Plain (IGP) compared with 2018 and 2019. However, the role of meteorological conditions and chemical reactions involving changes in air quality is not clear from these observation-based studies, which

only showed the phenomenon of concentration reduction and switch of major primary pollutants mainly in urban cities. Further, the number of monitoring stations in the country is way below the guidelines by the governing bodies and not uniformly distributed, which results in observation data limitations in India (Sahu et al., 2020).

In this study, the Community Multi-Scale Air Quality (CMAQ) model was used to investigate changes in air pollutants during the pre-lockdown (from February 21, 2020 to March 23, 2020) and lockdown (from March 24, 2020 to April 24, 2020) periods throughout Indian region. We explored the synergetic impacts from the meteorological conditions and anthropogenic emissions during the pre-lockdown and lockdown periods. Besides, we directly quantified the change in air quality during the lockdown due to the reduced anthropogenic emissions by comparing the differences between Case 1 (without emission reductions) and Case 2 (with emission reductions). The model performance was evaluated by comparing the simulation results with the observation data, which is collected by the Central Pollution Control Board (CPCB). This study has important implications for developing control strategies to improve air quality in India.

## 2 Methodology

### 2.1 Data collection

We used observed hourly $PM_{2.5}$, $O_3$, carbon monoxide (CO), and nitrogen dioxide ($NO_2$) data from February 21, 2020 to April 24, 2020 from the CPCB online database (https://app.cpcbccr.com/ccr/#/caaqm-dashboard-all/caaqm-landing, last access: July 07, 2020), which is widely applied in previous studies (Kumar, 2020; Sharma et al., 2020; Srivastava et al., 2020; Shehzad et al., 2020). The CPCB database provides data quality assurance (QA) or quality control (QC) programs by establishing strict procedures for sampling, analysis, and calibration (Gurjar et al., 2016). Besides, the observed daily averages of $PM_{2.5}$ and maximum daily 8-h average ozone (MDA8 $O_3$) have been further calculated to analyze the change in air quality during the pre-lockdown (from February 21, 2020 to March 23, 2020) and lockdown (from March 24, 2020 to April 24, 2020). The satellite-observed $NO_2$ and formaldehyde (HCHO) column number density datasets are from the Sentinel-5 Precursor TROPOspheric Monitoring Instrument (S-5P TROPOMI) (https://scihub.copernicus.eu). Besides, we effectively removed the pixels with a QA value less than 0.75 for $NO_2$ tropospheric column density and 0.5 for HCHO from the datasets to exclude the interferences such as clouds and snow/ice (Apituley, 2018).

### 2.2 Model description

This study applied CMAQ (Byun and Schere, 2006) version 5.0.2 with updated SAPRC-11 photochemical mechanism (Carter, 2011; Hu et al., 2016) and aerosol module (AERO6) (Binkowski and Roselle, 2003) to simulate air pollution across India with a horizontal resolution of 36 km × 36 km (117 × 117 grid cells). Figure 1 shows the simulation domain with positions of main Indian cities. The simulation was conducted from February 21 to March 23 as a pre-lockdown and March 24 to April 24 as a lockdown period.

The Weather Research & Forecasting model (WRF) version 3.6.1 was utilized to generate meteorology fields driven by the latest FNL (Final) Operational Global Analysis data. Anthropogenic emissions were from the monthly data from the Emissions Database for Global Atmospheric Research (EDGAR) version 4.3 (http://edgar.jrc.ec.europa.eu/overview.php?v=431). The monthly emissions from different source sectors were divided into six major groups of residential, industrial, agriculture, on-road, off-road, and energy before being adjusted from the base year of 2010 to 2019 based on population and economic growths similar to Guo et al. (2017) and the adjustment factors are shown in Table S1-S3. Weekly and diurnal profiles were used to convert monthly emissions to hourly inputs and the US EPA's SPECIATE 4.3 source profiles were used to speciate total particulate matters (PM) and volatile organic compounds (VOCs) to model species (Wang et al., 2014).

The biogenic emissions were derived from The Model of Emissions of Gases and Aerosols from Nature (MEGAN) version 2.1 (Guenther et al., 2012), and the emissions from biomass burning for 2018 were based on the Fire Inventory from the National Center for Atmospheric Research (FINN) (Wiedinmyer et al., 2011).

## 2.3 Emission reduction during COVID-19

Due to the COVID-19 lockdown, human activities were limited and related anthropogenic emissions were reduced. Different sources were used to obtain changes in anthropogenic emissions from different sectors in comparison to 2019.

For the sector of on-road and off-road, the vehicle emissions changes were based on the number of registered vehicles verified from the article (Bureau, 2020). The changes in energy demand were obtained from official data released by Power System Operation Corporation (POSOCO) (Abdi, 2020). Residential and agricultural emissions remain unchanged due to a lack of sufficient information.

For the industrial sector, we classify the Indian industries into 3 different classes based on the degree of air pollution caused (https://www.indianmirror.com/indian-industries/environment.html) (Table S4) including very polluting (VP), medium polluting (MP), and low polluting (LP) industries. The Pollution Index (PI) of any industry is a number ranging from 0 to 100 and the increasing value of PI denotes the increasing degree of pollution load from the industry. Besides, CPCB, State Pollution Control Boards (SPCBs), and the Ministry of Environment, Forest and Climate Change (MoEFCC) have finalized the criteria on the range of PI for the purpose of categorization of the industrial sector (https://pib.gov.in/newsite/printrelease.aspx?relid=137373) (Table 1).

Based on the above definition of the VP, MP, and LP industry, the emissions before lockdown can be expressed as:

$$E_1 = N_{VP\text{-}pre} \times S_{VP} + N_{MP\text{-}pre} \times S_{MP} + N_{LP\text{-}pre} \times S_{LP} , \tag{1}$$

where $S_{VP}$, $S_{MP}$, and $S_{LP}$ are 1, 0.6, and 0.4 as the assigned scores, and $N_{VP\text{-}pre}$, $N_{MP\text{-}pre}$, and $N_{LP\text{-}pre}$ are the number of each category industry during pre-lockdown. Similarly, the emissions during the lockdown are as follows:

$$E_2 = N_{VP\text{-}lock} \times S_{VP} + N_{MP\text{-}lock} \times S_{MP} + N_{LP\text{-}lock} \times S_{LP} , \tag{2}$$

where $N_{VP\text{-}lock}$, $N_{MP\text{-}lock}$, and $N_{LP\text{-}lock}$ are the number of functioning industries during the lockdown. Therefore, the percent reduction of industrial emissions can be calculated as:

$$\%reduction = \frac{E_1 - E_2}{E_1} \times 100 \, , \tag{3}$$

In this study, two sensitivity simulations were conducted during the lockdown periods. Case 1 assumes business as usual with the same emissions as in 2019, while Case 2 adjusts anthropogenic emissions using factors obtained above for different sectors (Table 2). The differences between Case 2 and Case 1 can be assumed as the effects of COVID-19 lockdowns.

## 3 Results and discussion

### 3.1 WRF-CMAQ model validation

Meteorology plays an important role in emissions, transport, deposition, and formation of air pollutants (Zhang et al., 2015). Hence, the performance of WRF is validated to assure accurate air pollution simulation against available observation from the National Climate Data Center (NCDC). There are more than 1300 stations within the simulation domain with hourly observations. The considered variables contain temperature at 2 m above the surface (T2), wind speed (WS), wind direction (WD), and relative humidity (RH). Table S5 shows the statistics of mean observation and mean prediction of meteorological parameters, along with mean bias (MB), gross error (GE), and root mean squared error (RMSE), which are compared to benchmarks suggested by Emery et al. (2001a). All the statistical indexes are listed in Table S6.

In general, the WRF model performance is similar to previous studies in India (Kota et al., 2018). For the pre-lockdown and lockdown period, predicted T2 was under-estimated with MB values of -1.5 K and -1.2 K, respectively. The GE values for WS were 1.7% (pre-lockdown) and 1.8% (lockdown), satisfying the suggested criteria of 2.0%, and RMSE was slightly over the criteria. The MB values for WD were 3.2° and 2.6° during the two periods, which are within the criteria of ±10°. The GE and RMSE for WD were slightly out of the benchmarks. The under-predicted RH was also observed in this study, which was reported in other Asian studies (Hu et al., 2015). Those statistic values that did fall in the benchmark were mainly due to the resolution (36 km) applied in this study compared to the finer resolution (4–12 km) suggested in Emery et al. (2001b) (Sahu et al., 2020).

Table S7 shows the model performance of MDA8 $O_3$, $PM_{2.5}$, CO, and $NO_2$ in five major cities in India including Delhi, Mumbai, Chennai, Hyderabad, and Bengaluru. For $PM_{2.5}$, after excluding some abnormally high values of greater than 300 µg $m^{-3}$, the averaged mean fractional bias (MFB) (-0.48) and mean fractional error (MFE) (0.61) values in all the five urban cites met the criteria limits of ±0.6 and 0.75 claimed by the EPA (2007). And the recommended criteria are commonly used for validating air quality model performance in the Indian region (Mohan and Gupta, 2018; Kota et al., 2018). For $O_3$, a cut-off value of 40 ppb is applied, which is based on EPA's recommendations (EPA, 2005). Besides, the model was able to reproduce the variation trends of observed hourly $O_3$ in all these major cities, although slightly over-estimations have occurred. And averaged MFB (-0.05) and MFE (0.25) values of $O_3$ also satisfy the benchmarks of ±0.15 and 0.30 set by the EPA (2005) in most of these cities with Chennai and Hyderabad exceeding the limits slightly. The performance of $PM_{2.5}$, $NO_2$, $O_3$, and CO in these urban areas was also similar to Kota et al. (2018), which could provide robust results for the following air quality study.

To further validate modeled HCHO and $NO_2$, we compared our simulated results with satellite-observed data during pre-lockdown and lockdown periods (Fig. S1). The CMAQ predicted vertical column densities (VCD) of tropospheric $NO_2$ and HCHO were calculated using Eq. (4) (H. J. Eskes, 2020).

$$VCD=\sum_{i=1}^{n} C_i \times H_i \times \alpha , \tag{4}$$

where n equals 17 as the number of vertical layers in the model (with the highest layer height of ~10 km), $C_i$ means species concentration (ppm), $H_i$ represents each layer height (m), and $\alpha$ is the coefficient for converting units from ppm to molec $cm^{-2}$. The predicted regional distribution of tropospheric column $NO_2$ and HCHO is similar to satellite-observations. Overall, HCHO and $NO_2$ are higher in eastern and northern India than in other regions. And their variation trends from CMAQ and TROPOMI are consistent that $NO_2$ decreases while HCHO increases during the lockdown. We also acknowledge that the uncertainty of emission inventory and chemical mechanism in the modelling may affect the simulated results (Dominutti et al., 2020; Kitayama et al., 2019).

## 3.2 Changes in air quality from pre-lockdown to lockdown periods

Figure 2 shows predicted and observed $PM_{2.5}$ from February 21 to April 24 in Delhi, Mumbai, Chennai, Hyderabad, and Bengaluru. The model succeeds in estimating the observed peak and valley values with slight under-estimation in all these cities. Overall, sharp decreases are found in the observed $PM_{2.5}$ in all these cities, and the averaged $PM_{2.5}$ level drops from 43.18 $\mu g\ m^{-3}$ to 27.62 $\mu g\ m^{-3}$. The mean observed $PM_{2.5}$ concentrations during lockdown are 42.47 $\mu g\ m^{-3}$ (Delhi), 24.53 $\mu g\ m^{-3}$ (Mumbai), 15.73 $\mu g\ m^{-3}$ (Chennai), 31.29 $\mu g\ m^{-3}$ (Hyderabad), 24.08 $\mu g\ m^{-3}$ (Bengaluru), which are reduced by 41%, 40%, 42%, 10%, and 43% respectively compared with that of the pre-lockdown period. Besides, the observed peak values of $PM_{2.5}$ in each city also decrease appreciably (up to 57%) during the lockdown period. On March 24 that the first day of lockdown, a significant drop in $PM_{2.5}$ concentration due to the emission reduction of primary pollutants is observed (Fig. S2). However, most of the $PM_{2.5}$ concentrations are still above the WHO annual guideline values of 10 $\mu g\ m^{-3}$ (WHO, 2016) during the lockdown period, with peak values over 60 $\mu g\ m^{-3}$ occasionally.

Figure 3 shows the temporal variation of MDA8 $O_3$ in these five cities. The predicted MDA8 $O_3$ is consistent in trend with observation values in most days, while simulated concentrations are overall higher, particularly in Hyderabad. The observed average MDA8 $O_3$ during lockdown is higher than that of pre-lockdown in Delhi (2%), Hyderabad (12%), and Bengaluru (2%). This is likely due to the fact that $O_3$ formation in these cities is under VOC control (Sharma et al., 2020), and nitrogen oxide ($NO_x$) reduction leads to $O_3$ increase by enhanced hydrogen oxide radicals ($HO_x$) concentrations (Zhao et al., 2017). The increase of monthly average T2 from pre-lockdown (281.0 K) to lockdown (285.1 K) could also lead to an increase of $O_3$ (Chen et al., 2019). In contrast, the observed average MDA8 $O_3$ during lockdown is reduced compared with the pre-lockdown period in both Mumbai (-35%) and Chennai (-13%). This could be caused by a much larger reduction in emissions as Mumbai and Chennai with high urbanization and industrialization are the most affected areas. In specific, more stringent lockdown measures may be implemented in Mumbai than we assumed, which accounted for more than a fifth of infections in India (Mukherjee, 2020).

Figure 4 shows the comparison of predicted air pollutants before and during the lockdown throughout India. Generally, decreases of key pollutants including particulate matter with an aerodynamic diameter of less than 10 μm ($PM_{10}$) (-16%), $PM_{2.5}$ (-26%), MDA8 $O_3$ (-11%), $NO_2$ (-50%), and sulfur dioxide ($SO_2$) (-14%) are calculated across India. Changes in these

195 pollutants present distinct regional variations. In northern and western India, the lower levels of these pollutants are observed during the lockdown, with the reductions of $PM_{2.5}$ and $PM_{10}$ up to 79%. In particular, the most significant decreases are found in the populated, industrialized, and polluted IGP region during the lockdown. The average $PM_{2.5}$ even drops from approximately 35–70 μg m$^{-3}$ (pre-lockdown) to 15–40 μg m$^{-3}$ (lockdown) in these regions because local emissions are generally the largest contributor (38–78%) to $PM_{2.5}$ in India (David et al., 2019). However, increases in these key pollutants are found

mainly in the northeastern, eastern, and parts of southern India.

Besides, changes in $PM_{2.5}$ also show prominent differences in the rural and urban areas. In India, rural areas have different emission sources from urban areas and are less influenced by lockdown measures (Garaga et al., 2020). In megacities such as Delhi, the predicted concentrations of $PM_{2.5}$ decline during the lockdown, which is consistent with previous results (Kumari and Toshniwal, 2020; Chauhan and Singh, 2020). For instance, over a 60% reduction of $PM_{2.5}$ is estimated in Delhi and

205 Ahmedabad. However, increases of $PM_{2.5}$ (~20%) are observed in the far-flung northeastern part of India. Variations in near-surface meteorological factors during lockdown also play an important role in $PM_{2.5}$ changes. As is shown in Fig. S3, lower $PM_{2.5}$ in urban areas during lockdown (Fig. 4) may be attributed to the decrease of RH and increase of planetary boundary layer (PBL) height, while the decrease of precipitation and WS allows $PM_{2.5}$ to accumulate in some rural areas (Schnell et al., 2018; Le et al., 2020).

As gaseous precursors of major components to $PM_{2.5}$ (Jain et al., 2020), concentrations of $NO_2$ and $SO_2$ also decrease significantly in most regions by up to 90% and 87%, respectively. However, their levels increase in parts of the east and south India and thus leading to higher levels of $PM_{2.5}$ and $PM_{10}$ in the same regions. MDA8 $O_3$ is also rising in eastern India by the highest increasing rate of 29%, while a 30% reduction is observed in northern and western India. Although significant reductions are found in $O_3$ precursor emissions throughout India during the lockdown, the MDA8 $O_3$ has not shown a

comparable decrease, which is affected by meteorological conditions such as an increase of temperature and decrease of RH (Fig. S3). Higher temperature speeds up photochemical processes that produce $O_3$, while higher RH reduces them (Chen et al., 2019; Zhao et al., 2017; Ali et al., 2012).

In summary, the decrease of $PM_{2.5}$, $PM_{10}$, $NO_2$, $SO_2$, and the increase of MDA8 $O_3$ during lockdown is consistent with previous results (Srivastava et al., 2020; Mahato et al., 2020). In the case of Delhi, compared with the previous studies, the $PM_{2.5}$

reduction (34%) is comparable with 35% reported by Chauhan and Singh (2020), while less than 53% stated by Mahato et al. (2020) and 49% calculated by Kumari and Toshniwal (2020) during the first phase of lockdown (from March 24, 2020 to April 15, 2020). These differences may be caused by the considered duration of the lockdown period. The later lockdown period (after April 15, 2020) is concerned in our study when there is an increase in traffic flow and some relaxation of lockdown measures (Kumar, 2020). Moreover, the different characteristics of these air pollutants in rural and urban areas have not been

investigated comprehensively in previous studies. Kumari and Toshniwal (2020) also concluded that concentrations of $PM_{10}$,

PM$_{2.5}$ and SO$_2$ tended to rise in Singrauli (rural area, located in central India) during the lockdown, contrary to the results of Delhi and Mumbai. Therefore, our results have important implications for the study of air quality changes and their regional distribution across India and indicate more strident emission reduction policies should be implemented across India, especially in the later phases of lockdown and in rural areas.

### 3.3 Effects of emission reductions on PM$_{2.5}$ during the lockdown

There are significant changes in PM$_{2.5}$ between the lockdown and pre-lockdown periods. Moreover, we directly quantify the change in PM$_{2.5}$ during the lockdown. Figure 5 shows the differences in major PM$_{2.5}$ components during the lockdown period with (Case 2) and without (Case 1) control measures.

Major components of PM$_{2.5}$ including nitrate (NO$_3^-$), sulfate (SO$_4^{2-}$), ammonium (NH$_4^+$), elemental carbon (EC), primary organic aerosol (POA), and secondary organic aerosol (SOA), decreased significantly in Case 2 compared to Case 1, indicating the positive effects of emission reduction. Primary components of PM$_{2.5}$ (EC and POA) are lowered by an average of 37% and 14%, respectively. EC is usually emitted from combustion sources and a drastic decrease of up to 74% directly reflected the impact of emission reductions from industry and transportation. Secondary inorganic aerosol (SIA) including NO$_3^-$, SO$_4^{2-}$, and NH$_4^+$ and SOA accounted for most of the PM$_{2.5}$ bulk mass (39%) and showed greater decreases than primary components. Moreover, the spatial distribution of SIA is similar to PM$_{2.5}$ in that the reduction is more significant in the north of India where the decrease of NO$_3^-$, SO$_4^{2-}$, and NH$_4^+$ are up to 92%, 57%, and 79% respectively. The largest reduction of NO$_3^-$ by averaged 62% resulted from transportation reduction and SO$_4^{2-}$ reduction (averaged 31%) is likely due to the falling release of industry (Gawhane et al., 2017; Wang et al., 2020). On average, NH$_4^+$ and SOA are decreased by 41% and 14%, respectively. The significant decrease in NH$_4^+$ cannot be attributed to the absence of reduced agricultural emissions in the simulation but may be due to the relatively reduced (NH$_4$)$_2$SO$_4$ and NH$_4$NO$_3$ in the CMAQ chemistry-transport model (Fountoukis and Nenes, 2007). By contrast, compared with VOCs, an important precursor of SOA, the smaller reduction of SOA may be related to the weakening of the atmospheric oxidizing capacity (AOC), which plays an important role in the formation of SOA (Feng et al., 2019). Besides, the reduction of NO$_x$ may lead to an increase in SOA offsetting some of the influence by the reduction in VOC emissions (Kroll et al., 2020).

Figure 6 shows the predicted response of changes in concentration of primary PM$_{2.5}$ (PPM) and secondary components to the reduced emissions of related precursors in Delhi, Mumbai, Kolkata, Bengaluru, Hyderabad, Chennai, Ahmedabad, and Lucknow. Generally, all species decreased with the reduced emissions and the great sensitivity of PM$_{2.5}$ component concentrations to emissions showed the important role of meteorology and the effectiveness of stringent measures to reduce emissions.

On average, NO$_3^-$ shares the largest reduction of 77% mainly driven by the decrease of its gaseous precursor NO$_x$ (71%). At least a 27% decrease of SO$_4^{2-}$ is found in each city caused by the largest reduction of SO$_2$ (averaged 59%). Over 70% average reduction of NO$_x$ and NO$_3^-$ may still relate to the reduction of vehicles. And SOA is dropped by an average of 18% because of the lack of precursors due to the emission reduction of VOCs (29%). Due to the reduction of emitting precursors, the

concentration reduction of PM$_{2.5}$ secondary components is less than that of primary components. The ratios of PPM reduction in emission (averaged 39%) are larger than the reduction in concentration (averaged 43%) in five selected cities. Especially, a 7% reduction in emission of PPM caused a 43% decline in its concentration in Hyderabad. Emissions of EC and organic carbon (OC) have also been reduced by a certain proportion resulting in a similar or greater reduction in concentrations.

The response of concentration to emissions in all cities presented a nonlinear change that has been confirmed previously by Zhao et al. (2017), which is related to various meteorological conditions (Wang et al., 2020). For example, in Lucknow, PPM, EC, OC, SO$_2$, NO$_x$, and VOCs decreased by 14%, 25%, 8%, 39%, 55%, and 11% respectively, while the concentration of PPM, EC, OC, SO$_4^{2-}$, NO$_3^-$, and SOA dropped by 21%, 32%, 12%, 43%, 78%, and 18%. Besides, the concentration response to emission reduction is likely to be more prominent in highly polluted and industrialized areas. The highest reductions in PPM and these secondary components of PM$_{2.5}$ happened in Ahmedabad (an industrial city located in western India) with high vehicular populations. While Bengaluru, a major southern Indian city, is considered as one of the cleaner Indian major cities because of its low PM$_{2.5}$ concentrations with no heavy industries (Guttikunda et al., 2019). Consequently, the reduction in PM$_{2.5}$ and its major components (especially for secondary components) in Bengaluru is not as significant as in Ahmedabad although a similar reduction in emissions is observed.

### 3.4 Effects of emission reductions on O$_3$ during the lockdown

We investigated the changes of MDA8 O$_3$ and its major precursors NO$_x$ and HCHO during the lockdown period. HCHO is one of the major contributors to total VOCs reactivity (Zhang et al., 2012; Steiner et al., 2008). It also has a strong correlation with VOC ($R^2$ up to 0.93) (Fig. S4) and performs well when validated by comparing with satellite-observed data. As a result, HCHO is used as a good proxy in the model for the total VOCs, consistent with previous studies such as Palmer et al. (2003). Figure 7 shows that MDA8 O$_3$, NO$_x$, and HCHO decreased all over India. The average reduction rates of MDA8 O$_3$, NO$_x$, and HCHO are approximately 15%, 50%, and 15%, respectively. For both Case 1 and Case 2, the higher levels of MDA8 O$_3$ are in eastern India (over 60 ppb, Case 1) in which the higher NO$_x$ is also observed (over 12 ppb, Case 1) during the lockdown. Compared to PM$_{2.5}$, no significant north-south differences are found in the change of O$_3$. NO$_x$ concentration has the greatest reduction that is mostly driven by the large cutting of energy emission by 26%, which is consistent with the decline of India's electricity consumption (9.2%) (Reuters, 2020).

Figure S5 shows the O$_3$ production sensitivity (O$_3$/NO$_y$) in India during the lockdown, which is considered as an indicator of O$_3$ sensitivity to NO$_x$ and VOCs (Sillman, 1995; Sillman and He, 2002). Besides, O$_3$/NO$_y$ < 6 indicates that O$_3$ formation is VOC-limited, O$_3$/NO$_y$ > 8 indicates NO$_x$-limited, and intermediate values are transitional. In India, NO$_x$-limited regimes are found in vast areas from both Case 1 and Case 2, which was also reported in previous studies (Mahajan et al., 2015). As a result, the large reduction of NO$_x$ leads to decreased MDA8 O$_3$ in most Indian regions. Compared to Case 1, the VOC-limited area expands mainly in the northwest and south of India from Case 2 during the lockdown. Simultaneously, the rise of MDA8 O$_3$ (averaged 5% and up to 21%) is found sporadically in these VOC-limited areas in which more significant decreases of NO$_x$ (compared with VOCs) reduce the O$_3$ consumption (NO + O$_3$ = NO$_2$ + O$_2$) and enhance HO$_x$ concentrations result in an

increase in $O_3$ levels. It may also indicate that the increase in $O_3$ is amplified regionally by the expansion of the VOC-limited regimes due to the lockdown.

Figure 8 compares the concentrations of MDA8 $O_3$, HCHO, and $NO_x$ with emissions of VOCs, HCHO, and $NO_x$ in eight major cities of India, Delhi, Mumbai, Kolkata, Bengaluru, Hyderabad, Chennai, Ahmedabad, and Lucknow. Generally, the decline in $O_3$ concentration in Delhi (14%), Mumbai (23%), Kolkata (24%), Bengaluru (20%), Hyderabad (17%), Chennai (20%), Ahmedabad (21%), and Lucknow (15%) showed that effectiveness of emission reductions that play an important role in the control of $O_3$ pollution, even in these VOC-limited areas.

The changes in emissions and concentrations of MDA8 $O_3$, HCHO, and $NO_x$ showed a non-linear response. In Delhi, a 76% reduction in $NO_x$ emissions resulted in a 77% reduction in its concentration, while a 29% reduction in HCHO resulted in only an 11% reduction. In a megacity like Delhi, about 7 million vehicles and many fossil fuel-based plants lead to high $NO_x$ emissions, and local restricted transportation and industrial activities during lockdown could lead to a significant reduction of primary $NO_x$ emissions (Sharma et al., 2016). The concentration of $NO_x$ is appreciably highly sensitive to a primary $NO_x$ emission reduction. However, the VOCs emission reduction resulting from the lockdown is relatively less than $NO_x$ in each city. And most of the reduction of HCHO concentration is less than that of emission reduction, which is different from $NO_x$, which indicated that the change of HCHO concentrations is not dominated by primary HCHO emission reduction.

## 4 Conclusion

Compared with pre-lockdown, observed $PM_{2.5}$ during the lockdown in Delhi, Mumbai, Chennai, Hyderabad, and Bengaluru shows an overall decrease. In contrast, MDA8 $O_3$ increases in three of these cities. The comparison of predicted air pollutants across India before and during the lockdown shows distinct regional characteristics. The most significant reductions of $PM_{2.5}$ and $PM_{10}$ (up to 79%) are observed in most of northern and western India including all these megacities. However, increases of MDA8 $O_3$ (up to 29%) and other key pollutants are reported in northeastern, eastern, and parts of southern India covering most of the rural areas. Besides, it can be concluded that the synergetic impact from the meteorological conditions and anthropogenic emissions plays an important role in those increases from pre-lockdown to lockdown.

The drastic decline in $PM_{2.5}$ and its major components during the lockdown period in Case 2 compared with Case 1 shows the positive impacts of emission control measures, especially for SIA. During the lockdown, the decrease of MDA8 $O_3$ (averaged 15%) occurs in most regions in India, which is attributed to the lower emissions of $NO_x$ (48%) and VOCs (6%) that are precursors of $O_3$. Our results demonstrate that the strident emissions controls due to the lockdown have mitigated air pollution in India. However, more stringent mitigation measures are needed to achieve effective control of air pollution from secondary air pollutants and their components, particularly in rural areas. We also find the scattered increases in MDA8 $O_3$ (up to 21%) in some urban locations in the VOC-limited areas due to the emissions reduction. This indicates that a more localized control policy with the consideration of the $O_3$ sensitivity regime should be implemented in India to improve the air quality especially for secondary pollutants such as $O_3$.

*Data availability.* The datasets used in the study can be accessed from websites listed in the references or by contacting the corresponding authors (peng.ce.wang@polyu.edu.hk; zhanghl@fudan.edu.cn).

*Author contribution.* MZ conducted the modelling and led the writing of the manuscript. AK carried out the data collection and initial analysis. SZ, JS, and JM assisted with the data analysis. MX, SK assisted with the interpretation of the results and the writing of the paper. HZ and PW designed the study, discussed the results, and edited the paper.

*Competing interests.* The authors declare that they have no conflict of interest.

*Acknowledgments.* We acknowledge the publicly available WRF and CMAQ models that make this study possible. This project was funded by the Institute of Eco-Chongming (ECNU-IEC-202001).

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

**Table 1: The criteria on the "Range of Pollution Index" for the purpose of categorization of industrial sectors.**

| Categories* | Pollution Index score |
|---|---|
| Very polluting (VP) | ≥60 |
| Medium polluting (MP) | 41–59 |
| Low polluting (LP) | 21–40 |

Note: * VP, MP, and LP industries are also defined as the red, orange, and green categories of industrial sectors respectively, based on the Indian Ministry of Environment, Forest and Climate Change website (https://pib.gov.in/newsite/printrelease.aspx?relid=137373).

**Table 2: Percent reduction in anthropogenic emissions in India during COVID-19 lockdown.**

| Sector | %Reduction |
|---|---|
| Residential | 0 |
| Industrial | 82 |
| Agriculture | 0 |
| On-road | 85 |

| Off-road | 85 |
| Energy | 26 |

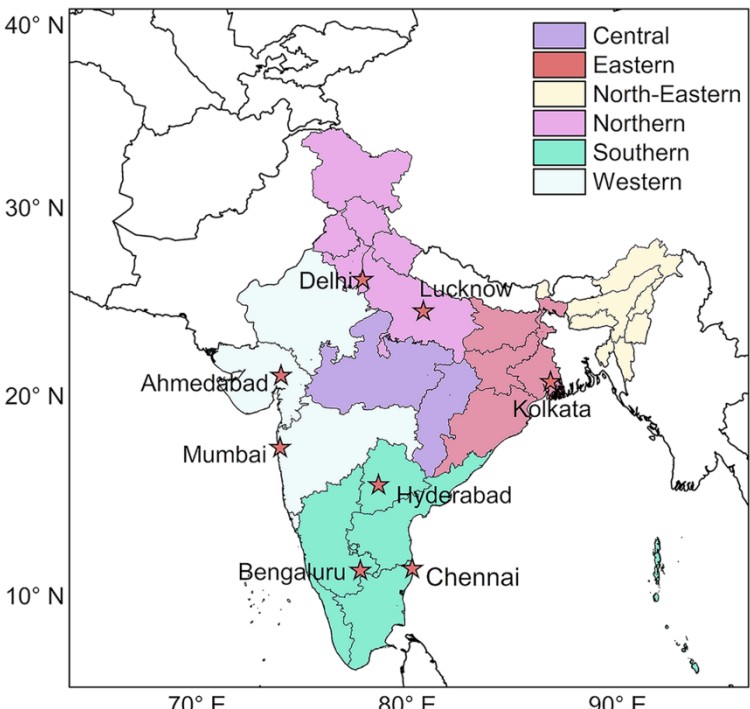

**Figure 1: The simulation domain with the location of major Indian cities selected for analysis.**

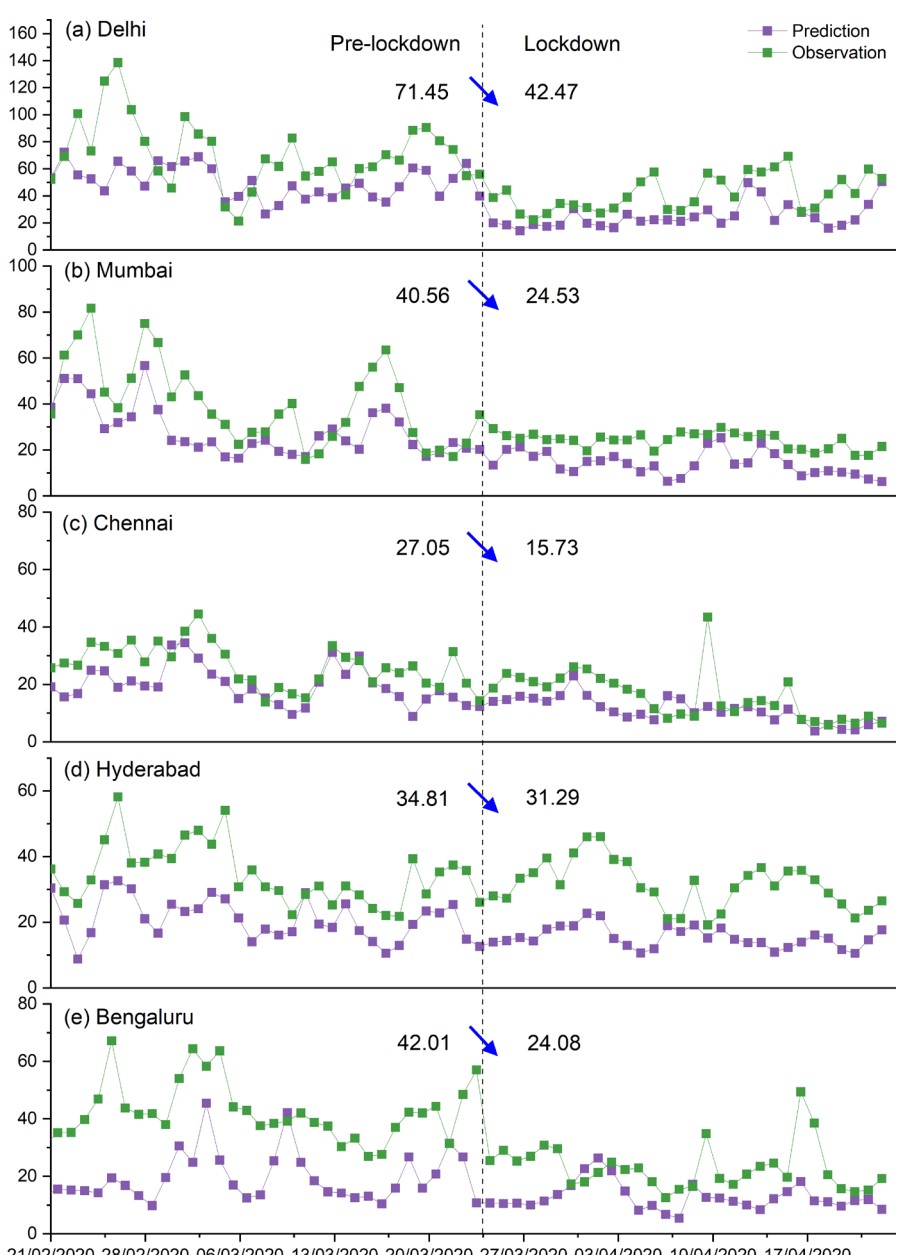

Figure 2: Comparison of predicted and observed PM$_{2.5}$ from February 21 to April 24, 2020 in Delhi, Mumbai, Chennai, Hyderabad, and Bengaluru. The unit is μg m$^{-3}$.

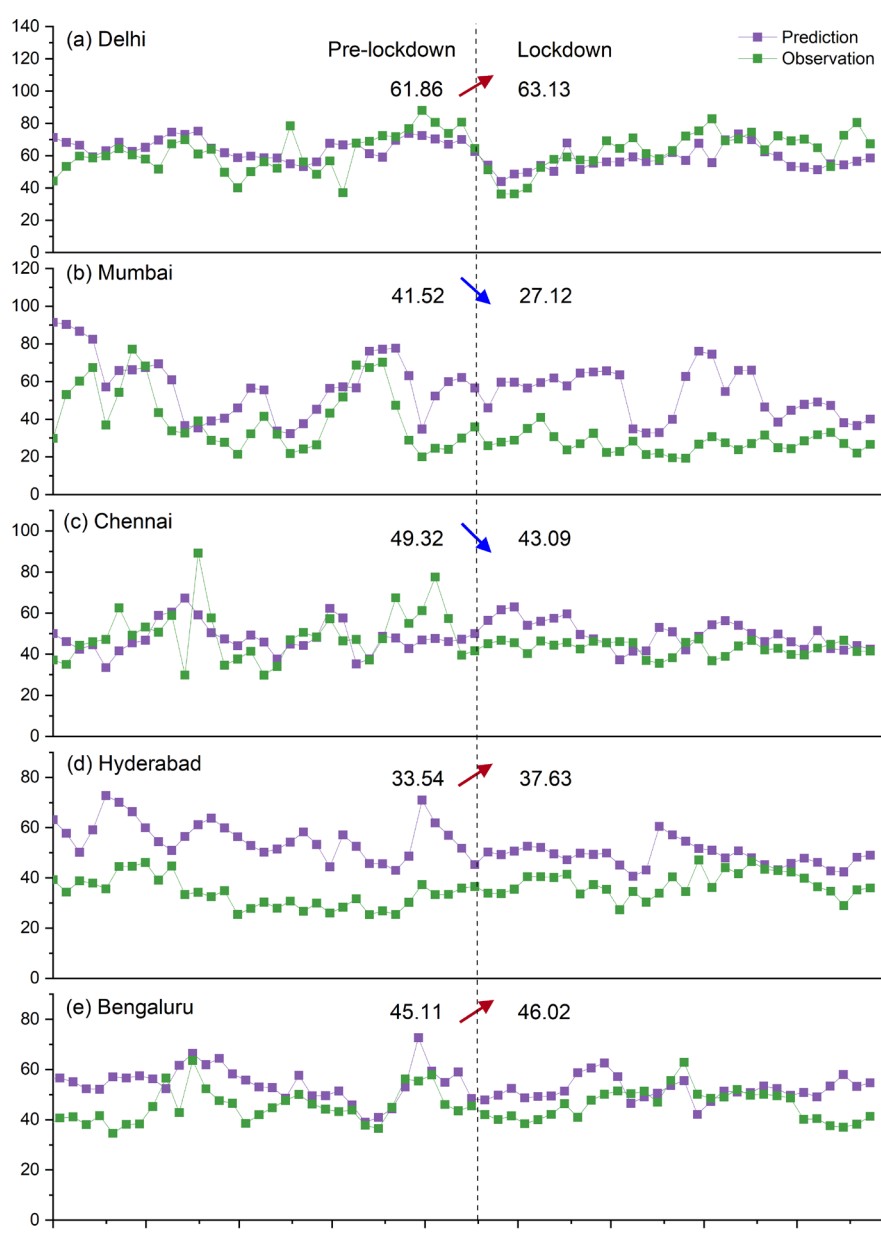

**Figure 3: Comparison of predicted and observed MDA8 O₃ from February 21 to April 24, 2020 in Delhi, Mumbai, Chennai, Hyderabad, and Bengaluru. The unit is ppb.**

555

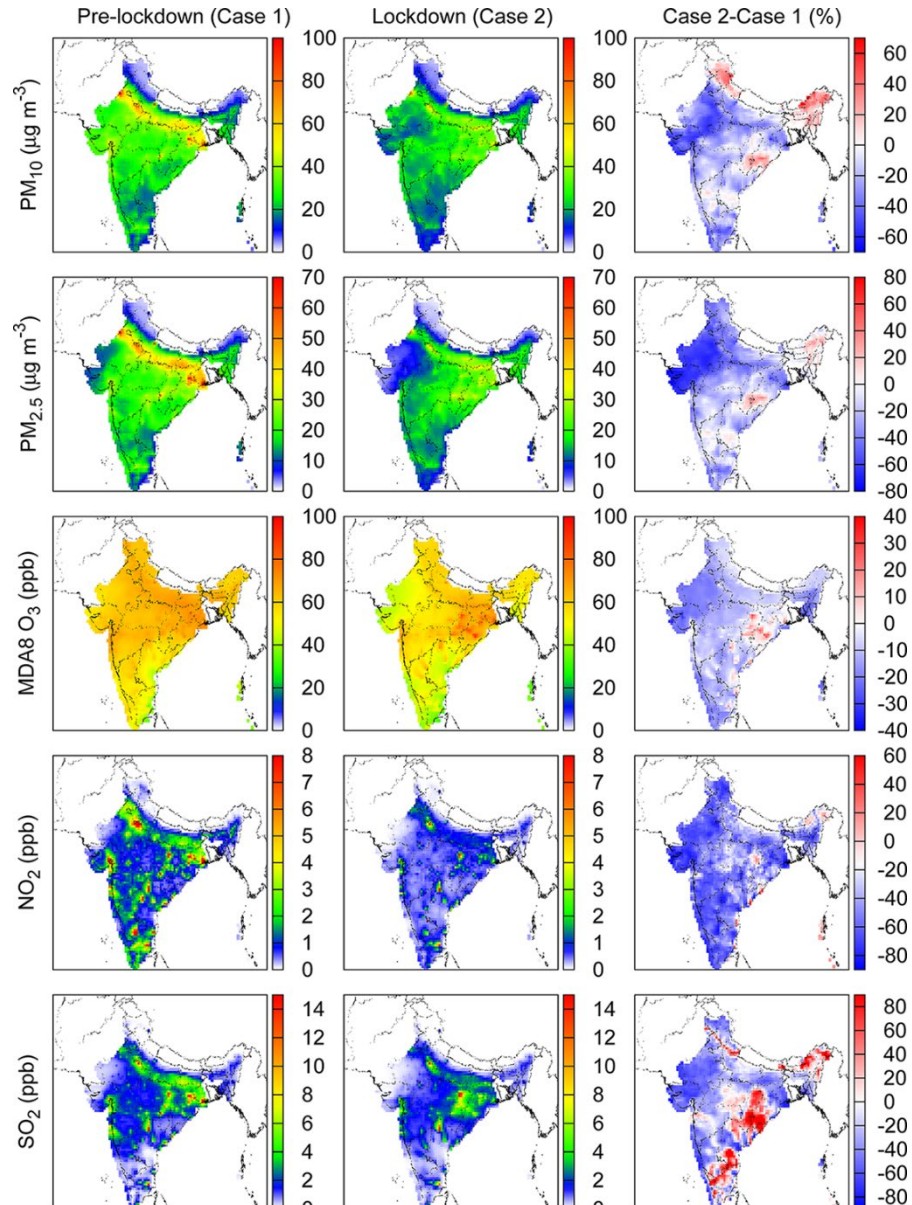

**Figure 4: Predicted PM₁₀ (µg m⁻³), PM₂.₅ (µg m⁻³), MDA8 O₃ (ppb), NO₂ (ppb), and SO₂ (ppb) before lockdown, during the lockdown and the changes between them in India. "Case2 - Case1" indicates (Case 2 – Case 1)/Case 1, reported as %.**

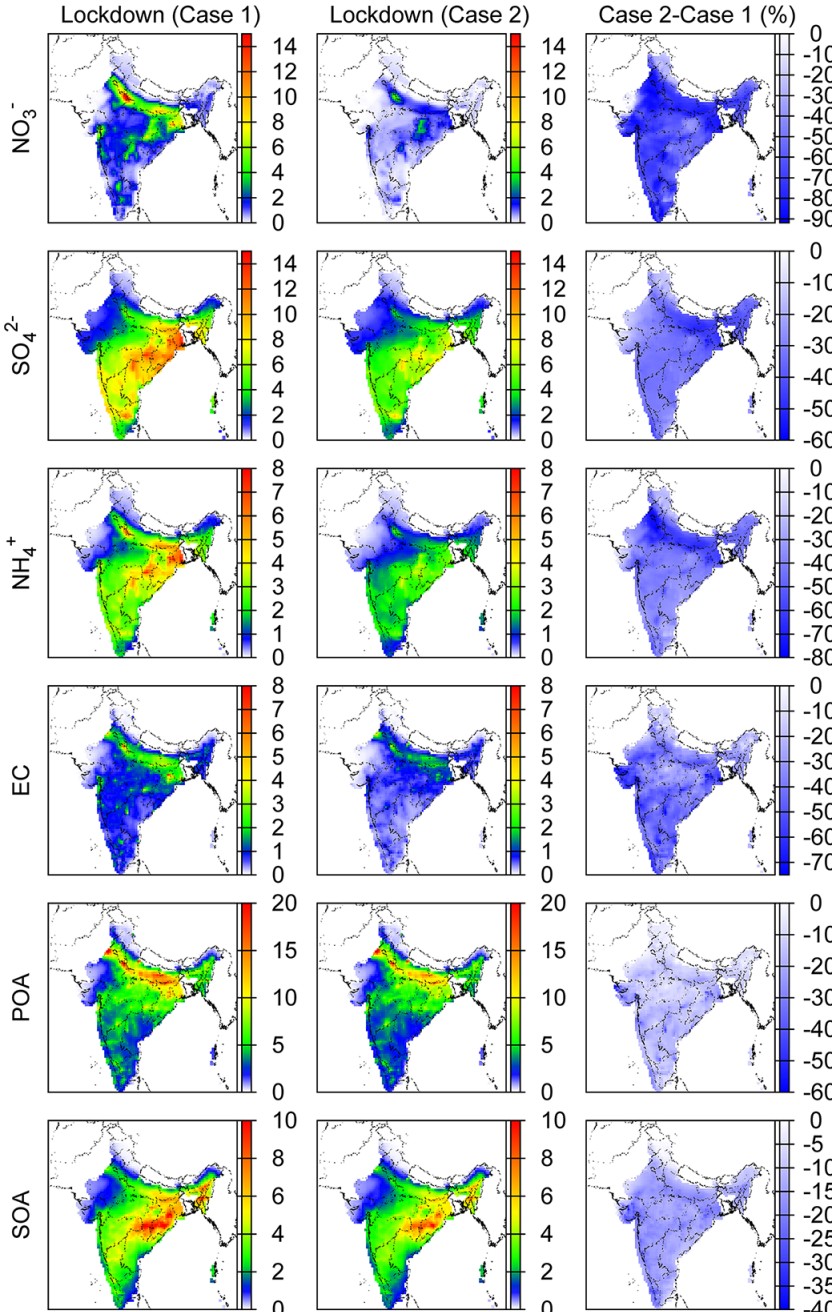

**Figure 5: Predicted PM$_{2.5}$ components and the changes caused by lockdown measures from March 24 to April 24, 2020 in India. The unit is μg m$^{-3}$. "Case2 - Case1" indicates (Case 2 – Case 1)/Case 1, reported as %.**

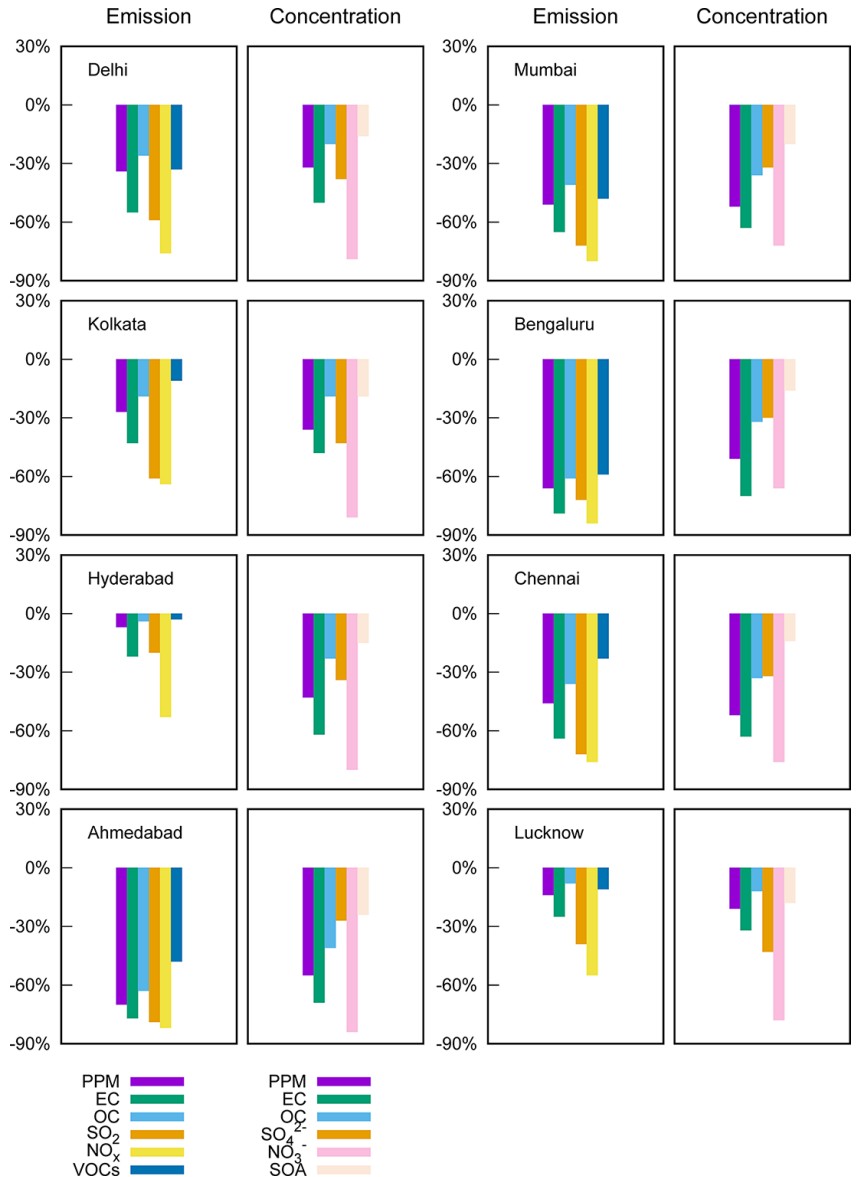

**Figure 6: Predicted relative changes in concentrations of primary and secondary components, and emissions of their precursors in eight cities of India in Case 2 to Case 1.**

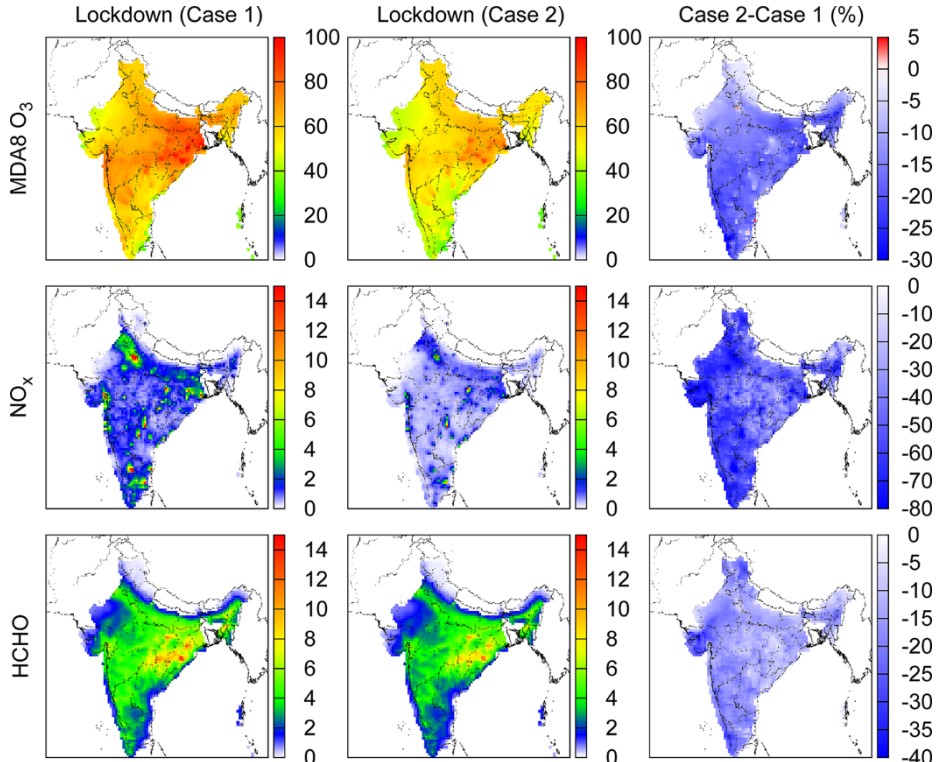

565

**Figure 7: Predicted O₃, NOₓ, HCHO, and the changes caused by nationwide lockdown measures from March 24 to April 24, 2020 in India. The unit is ppb. "Case2 - Case1" indicates (Case 2 – Case 1)/Case 1, reported as %.**

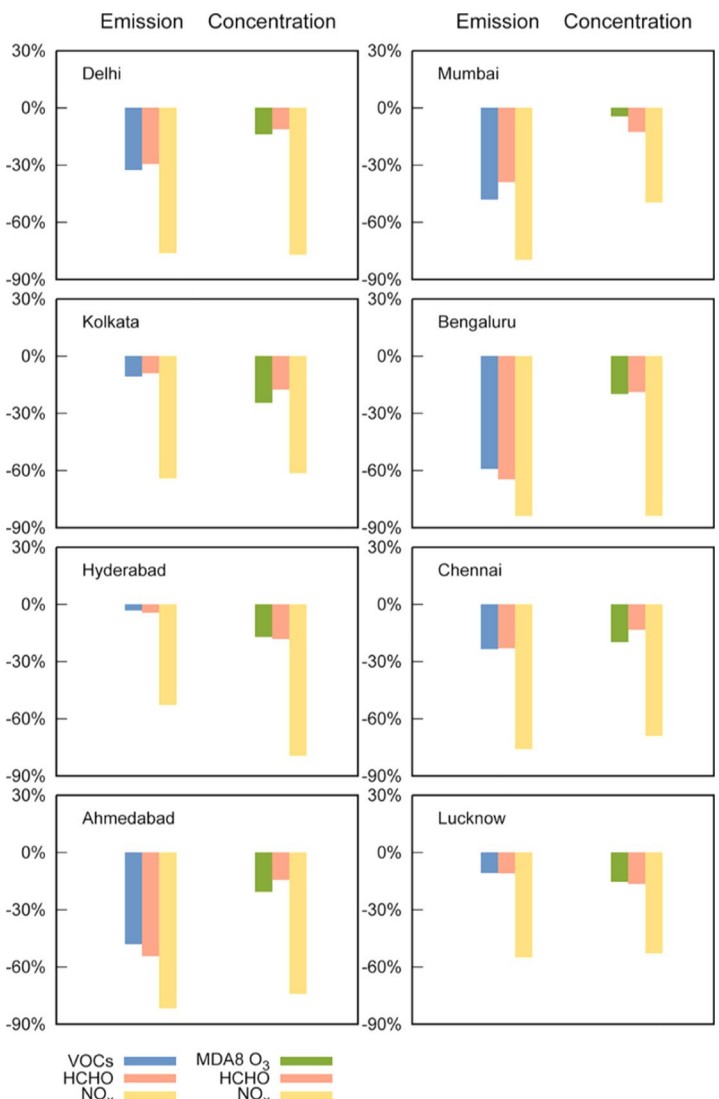

**Figure 8: Predicted relative changes in concentrations of O₃, HCHO, and NOₓ and emissions of VOCs, HCHO, and NOₓ in eight major cities of India in Case 2 to Case 1.**