# Peer review of "Impact of reduced anthropogenic emissions during COVID-19 on air quality in India"

_Atmospheric Chemistry and Physics, 2020_

## Short Comment (SC1) · 9 Oct 2020

The paper presents the percentage reduction in concentrations of air pollutants (PM2.5, O3, CO and NO2 etc.,) in India during lockdown period using ground observations (at Delhi, Lucknow, Kolkata, Ahmedabad, Mumbai, Hyderabad, Bangalore and Chennai during 21 February to 24 April 2020) and air quality models (WRF and CMAQ). The paper also described the percent reduction in secondary inorganic aerosols/species of PM2.5 over the region. A quick search on the web shows numerous studies on these lines published in the various reputed journals over Delhi and other locations of India. Some of them are cited in the present papers. A comprehensive study on the air quality of India during and before lockdown period has been carried by Kumar et al. (2020) in Sustainable Cities and Society (2020): 102382 and

they have also compared most of the recent studies.

The only advantage of this manuscript is that the authors have used the 7 stations ground based observation datasets (PM2.5, O3, CO and NO2 etc.,), validated the WRF and CMAQ models and extrapolated the pollutants for the country. This advantage gives the merit for publication of the paper in journal. However, the following points may also be considered before publication in ACP.

Authors are suggested to search all the recent articles published on these line in various journal over Indian and other region of the globe and make a summary Table and include in Introduction section. Also highlights how the present study is different from the previous studies as well new scientific information are going to be add in existing knowledge.

A number of publications are published recently on temporary reduction of concentrations of pollutants over India and other country during lockdown period. Hence, the present paper should be scientifically different and should add new information for scientific community.

Conclusion sections may be improved with avoiding the repetitive information from the abstract.

---

## Referee Comment (RC1) · Anonymous Referee #2 · 26 Oct 2020

The manuscript presents a topical research, i.e. to understand the air quality change in India during the COVID-19 lockdown period. The reported ozone and PM changes from the ground-based observations reveal the sensitivity of major pollutions to the drastic emission reduction in India which is one of the most polluted regions in the world. The WRF-CMAQ model simulations further shed light on the relative contributions from primary emissions and secondary formation of aerosols. The manuscript is easy to follow and fit to the scope of ACP very well. I recommend its publication with ACP, while I also have comments below for the authors to address.

1) The major results of this study are based on the comparison between the Lockdown and Pre-lockdown periods. Figs. 1-3 show observed changes in PM and ozone. I'm wondering how different the meteorological conditions are during those two periods?

The recent COVID-19-related studies (e.g. Le et al., 2020, Science) have stressed the importance of ventilation conditions and relative humidity in regulating the air pollution at the short time scale. Since the authors have obtain the meteorological data and conducted WRF simulations, it should be a low-hanging fruit to perform a more comparative analysis of meteorology (precipitation, winds, PBL height, etc.), in addition to the present Fig. S2 of temperature comparison. The outcome of this analysis should be also discussed in the context with the recent studies focusing on the air quality changes during the COVID-19.

2) There is no doubt that HCHO is closely linked with VOCs, but it is unclear to me how well the HCHO can be used to represent the total VOCs in a quantitative manner. Can the authors show their correlations based on the CMAQ model results?

3) Spaceborne measurements of HCHO and NO2 are available from the latest satellites, such as TROPOMI. There are near-real-time products. The authors may want to explore those data and validate the modeled HCHO and NO2 and see their changes in observations.

---

## Referee Comment (RC2) · Anonymous Referee #3 · 11 Nov 2020

**General Comments:**

The paper by Zhang et al. entitled "Impact of reduced anthropogenic emissions during COVID-19 on air quality in India" is on a very relevant and interesting topic which is to use the covid lockdown emission reductions for assessing impacts on air quality over India. Unfortunately, the analyses and interpretation are weak in several places (listed below). There are hardly any new trustworthy insights from this modelling study which have not been reported already by the authors in previous works on the same topic published recently (see Sharma, S., Zhang, M., Anshika, Gao, J., Zhang, H., and Kota, S. H.: Effect of restricted emissions during COVID-19 on air quality in India, Science of The Total Environment, 728, 138878, https://doi.org/10.1016/j.scitotenv.2020.138878, 2020). Instead there are even discrepancies from the earlier work based on interpretation of what appears to be the same measured dataset. While in the previous work (Sharma et al., 2020) it was reported that there was a 17% increase in ozone during COVID, in the present work it has been reported that a significant decrease in surface ozone (MDA8 values) occurred, without even clarifying what changed between the two studies except for additional modelling analyses in this study.

There are several major issues with the present submission which need to be addressed/clarified for meriting further publication in ACP.

**Validation of the model used in this work has not been done/described adequately:**

Authors use only 2 m level measurements of temperature and meteorological data and chemical data from 5 monitoring stations operated by the regulatory agency of India located within cities to compare their modelled output.

**Measured chemical data:** The authors present only daily averaged data in the plots (Figures 2 and 3). This would be fine but I could find no details of the original high resolution primary data (presumably available at temporal resolution of few minutes from the analyzers in the monitoring stations) to build confidence in the reader about the trustworthiness of the primary data and its quality assurance. If they could provide such high resolution data for the five stations (even for few days in both periods) for ozone , NO, NO2 , PM2.5 etc.. with gaps in measurements if any (after all there was a lockdown so maintenance could be difficult), and the calibration data of any of the analyzers, it would go a long way in instilling confidence in the highly averaged data. The reviewer looked up their previous study Sharma et al 2020 which has been cited for detailed description of the primary data and found that this reference did not contain these details and somewhat remarkably the Sharma et al. 2020 paper reported data until April 14th 2020 in that work, was submitted on April 16th , 2020 and accepted on April 19th , 2020. While this does not necessarily suggest that due diligence was not taken as given the nature of topic urgency to publish would have been a factor, the rapid turn-around time and lack of experimental details in the peer reviewed reference cited and which forms the basis of the daily averages does leave room for concern. So the authors should provide the original primary

data as a time series for these 5 monitoring stations in the revised supplement alongwith details of calibration experiments and data quality control followed to allay such potential concerns about the primary measured dataset.

Also they should discuss whether data from 5 cities are adequate to make inferences about all of India with same degree of confidence which spans vast rural and countryside regions ? It might be advisable to better focus on the 5 cities alone for which they have the data and even there they should acknowledge how data from one monitoring station may be limited for representing air quality of the entire city. In fact a combination of monitoring station data and satellite data (agreed also can have issues but better than nothing) would be better.

**Validation of model using 2 m level measured meteorological data:**

For the kind of modelling investigation the authors are making namely, effect of emission changes on concentrations of pollutants use of only the 2 m level observations without comparison with satellite data, sonde data, mixing layer height data (see ERA5 products) seems to be a major shortcoming. Note that the changes in ventilation coefficient before and during lockdown and the changing season (Spring to Summer) can alone have big impacts on the concentrations.

**Changes in atmospheric chemistry of primary pollutant removal and formation of secondary pollutants:**

Currently the study tends to attribute all the observed concentration changes in pollutants primarily to the emission reductions. However it has been documented elegantly in the following paper: Kroll, J.H., Heald, C.L., Cappa, C.D. et al. The complex chemical effects of COVID-19 shutdowns on air quality. Nat. Chem. 12, 777–779 (2020). https://doi.org/10.1038/s41557-020-0535-z, that several other processes play a big role.

The purpose of using a model should be that these effects can be teased out through sensitivity experiments but unfortunately this has not been addressed in current version of the manuscript. For example the authors note that the temperature increased during the lockdown period. A key question is what effect the temperature change and the reduced emission of VOCs (no VOC measurements have been provided at all), NOx and CO would have on the removal rates of primary pollutants and formation of secondary pollutants.

Further have authors identified days when it rained in both pre covid lockdown and during lockdown periods which would cause strong biases for the comparisons.

**Existing inadequacies in VOC emission inventories and modelled ozone simulations over India:** While the authors are using pre lockdown and lockdown periods for comparison, it is a fact that of all emission inventories, VOC emissions are the most poorly constrained due to the absence of in-situ VOC data over many regions in India. A generic problem also seen is the tendency for overestimation of ozone by models over the Indian region. This suggests that the basic reactant mixture and chemistry are still inadequate for modelling ozone and secondary pollutant formation accurately over India. So how can one be sure that the changed chemical mixture between pre-lockdown and during lockdown are not skewed by these gaps in our basic undertanding? While it would be unfair to hold the authors to solve all these issues, one does expect that the limitations and existing issues are duly acknowledged in the work instead of making highly speculative and prescriptive measures for air quality mitigation based on such modelling results. Use of formaldehye for constraining VOC emissions where a large number of more reactive primary VOC emissions occur should also be discussed and clarified. Trusting the formaldehye from the model in absence of in-situ formaldehye measurements to compare with or even satellite or columnar measurements which have been reported from India is recommended.

Are benzene and toluene data available from the monitoring stations which could be included in the analyses? If so these should also be included in view of their health and SOA formation potential.

**Choice of scaling factors for emission reductions:** The authors make several assumptions and justification for the use of scaling factors for emissions which are valid (see Equations 1 and 2).

For example:

**Ammonia agricultural emissions:** Several satellite studies have indicated high ammonia emissions from agriculture and a recent by G.K. Singh, P. Rajeev, D. Paul, et al., Chemical characterization and stable nitrogen isotope composition of nitrogenous component of ambient aerosols, Science of the Total Environment, https://doi.org/10.1016/j.scitotenv.2020.143032 showed that agriculture activities and waste generation are major sources of ammonia. The assumption by the authors that the agricultural emissions do not change between pre-lockdown and during lockdown is not valid for large parts of the India in particular the Indo-Gangetic Plain because during the pre-lockdown dates farmers were still applying fertilizers to the wheat crops, whereas by last week of March this completely stops. So infact the ammonia and hence ammonium ion source from agriculture is likely stronger in pre-lockdown period and so cannot be treated as constant between both periods. As ammonia is such an important emission for PM2.5 too, this has large implications for the inferences currently drawn by the authors.

**Ozone production sensitivity indicator:** The use of HCHO/NO2 as based on Silman et al 1995 which the authors cite cannot be applied blindly because as noted by the original authors (Silman and He in their JGR paper in 2002) is suitable only for ambient ozone mixing ratios in the range

of 80-200 ppb and then again for columns retrieved using satellite data. For ground based data, more robust proxies would be H2O2/ HNO3 or even O3/NOy . In the absence of measured VOC data presented by the authors to validate their model VOC data (note there are no measurements of HCHO presented), the authors should remove this discussion completely or present for each city site the high resolution O3 Vs NOx data from daytime for pre and during lockdown periods.

In several instances, the grammar and language also need to be corrected. I recommend the authors to consider the above major concerns to revise and improve the manuscript.

---

## Author Comment (AC1) · 25 Jan 2021

Dear Referee #2, We appreciate your comments to help improve the manuscript. We tried our best to address your comments and detailed responses and related changes are shown below. Our response is in blue and the modifications in the manuscript are in red. Besides, please note the supplementary PDF file in the reply.

Comments: The manuscript presents a topical research, i.e. to understand the air quality change in India during the COVID-19 lockdown period. The reported ozone and PM changes from the ground-based observations reveal the sensitivity of major pollutions to the drastic emission reduction in India which is one of the most polluted regions in the world. The WRF-CMAQ model simulations further shed light on the

relative contributions from primary emissions and secondary formation of aerosols. The manuscript is easy to follow and fit to the scope of ACP very well. I recommend its publication with ACP, while I also have comments below for the authors to address. Response: Thanks for the recognition of our study. Below is the response to each specific comment.

Comments: The major results of this study are based on the comparison between the Lockdown and Pre-lockdown periods. Figs. 1-3 show observed changes in PM and ozone. I'm wondering how different the meteorological conditions are during those two periods? The recent COVID-19-related studies (e.g. Le et al., 2020, Science) have stressed the importance of ventilation conditions and relative humidity in regulating the air pollution at the short time scale. Since the authors have obtain the meteorological data and conducted WRF simulations, it should be a low-hanging fruit to perform a more comparative analysis of meteorology (precipitation, winds, PBL height, etc.), in addition to the present Fig. S2 of temperature comparison. The outcome of this analysis should be also discussed in the context with the recent studies focusing on the air quality changes during the COVID-19. Response: Thanks for the comments. Considering the important influence of meteorological conditions, we added more analysis on their differences between pre-lockdown and lockdown periods including temperature (T), relative humidity (RH), planetary boundary layer (PBL) height, the average daily precipitation, and wind fields in below Fig. 1. We also explained the impacts of these meteorological conditions on air quality in the Results and Discussion section. Changes in manuscript: Results and discussion (Lines 199-203 in the revision): "Variations in near-surface meteorological factors during lockdown also play an important role in PM2.5 changes. As is shown in Fig. S3, lower PM2.5 in urban areas during lockdown (Fig. 4) may attribute to the decrease of RH and increase of planetary boundary layer (PBL) height, while the decrease of precipitation and WS allows PM2.5 to accumulate in some rural areas (Schnell et al., 2018; Le et al., 2020)." Results and discussion (Lines 207-211 in the revision): "Although significant reductions are found in O3 precursor emissions throughout India during the lockdown, the MDA8 O3 has

not shown comparable decreasing trends, which is affected by the meteorological conditions such as an increase of temperature and decrease of RH (Fig. S3). Higher temperature speeds up photochemical processes that produce O3, while higher RH reduces them (Chen et al., 2019; Zhao et al., 2017; Ali et al., 2012)."

Comments: There is no doubt that HCHO is closely linked with VOCs, but it is unclear to me how well the HCHO can be used to represent the total VOCs in a quantitative manner. Can the authors show their correlations based on the CMAQ model results? Response: As one of the most abundant oxygenated VOCs, HCHO is one of the major contributors to total VOCs reactivity (Zhang et al., 2012; Steiner et al., 2008). Therefore, it is used to show the model performance on VOCs due to the lack of VOCs observations. Figure 2 shows scatter plots comparing the simulated average daily HCHO and the total VOCs at all $117 \times 117$ grids during the study period. It can be seen from the results that HCHO has a high correlation with VOCs, and R2 reaches 0.93. We added explanations in the manuscript to make it clear. Changes in manuscript: Results and discussion (Lines 267-269 in the revision): "We investigated the changes of MDA8 O3 and its major precursors NOx and HCHO that is one of the major contributors to total VOCs reactivity (Zhang et al., 2012; Steiner et al., 2008) during the lockdown period. Figure S4 shows that HCHO has a strong correlation with total VOCs (R up to 0.93)."

Comments: Spaceborne measurements of HCHO and NO2 are available from the latest satellites, such as TROPOMI. There are near-real-time products. The authors may want to explore those data and validate the modeled HCHO and NO2 and see their changes in observations. Response: Thank you for this good suggestion. To validate the modeled HCHO and NO2, we compared our simulated results with satellite-observed NO2 and HCHO column number density datasets from TROPOMI during pre-lockdown and lockdown periods (Fig. 3). As shown in Fig. 3, the predicted regional distribution of NO2 and HCHO column number densities is similar to satellite-observations. Overall, HCHO and NO2 are higher in eastern

and northern India than in other regions. And their variation trends from CMAQ and TROPOMI are consistent that NO2 decreases while HCHO increases during the lockdown. Changes in manuscript: Methodology (Lines 85-89 in the revision): "The satellite-observed NO2 and formaldehyde (HCHO) column number density datasets are from the Sentinel-5 Precursor TROPOspheric Monitoring Instrument (S-5P TROPOMI) (https://scihub.copernicus.eu). Besides, we filter the satellite data under the recommended criteria of QA values greater than 75% for tropospheric NO2 column number density datasets and 50% for HCHO (Apituley, 2018)." Results and discussion (Lines 158-163 in the revision): "To further validate modeled HCHO and NO2, we compared our simulated results with satellite-observed data during pre-lockdown and lockdown periods (Fig. S1). The tropospheric column densities of NO2 and HCHO were calculated by summing their concentrations of 17 vertical layers in the CMAQ model (H. J. Eskes, 2020). The predicted regional distribution of tropospheric column NO2 and HCHO is similar to satellite-observations. Overall, HCHO and NO2 are higher in eastern and northern India than in other regions. And their variation trends from CMAQ and TROPOMI are consistent that NO2 decreases while HCHO increases during the lockdown."

Please also note the supplement to this comment:
https://acp.copernicus.org/preprints/acp-2020-903/acp-2020-903-AC1-supplement.pdf

[Figure]

**Fig. 1.**

[Figure]

**Fig. 2.**

[Figure]

**Fig. 3.**

**Supplement:**

**Responses to interactive comments**

**Journal: Atmospheric Chemistry and Physics**

**Manuscript ID: acp-2020-903**

**Title: "Impact of reduced anthropogenic emissions during COVID-19 on air quality in India"**

Dear Referee #2,

We appreciate your comments to help improve the manuscript. We tried our best to address your comments and detailed responses and related changes are shown below. Our response is in blue and the modifications in the manuscript are in red.

**Comments:** The manuscript presents a topical research, i.e. to understand the air quality change in India during the COVID-19 lockdown period. The reported ozone and PM changes from the ground-based observations reveal the sensitivity of major pollutions to the drastic emission reduction in India which is one of the most polluted regions in the world. The WRF-CMAQ model simulations further shed light on the relative contributions from primary emissions and secondary formation of aerosols. The manuscript is easy to follow and fit to the scope of ACP very well. I recommend its publication with ACP, while I also have comments below for the authors to address.

**Response**: Thanks for the recognition of our study. Below is the response to each specific comment.

**Comments:** The major results of this study are based on the comparison between the Lockdown and Pre-lockdown periods. Figs. 1-3 show observed changes in PM and ozone. I'm wondering how different the meteorological conditions are during those two periods? The recent COVID-19-related studies (e.g. Le et al., 2020, Science) have stressed the importance of ventilation conditions and relative humidity in regulating the air pollution at the short time scale. Since the authors have obtain the meteorological data and conducted WRF simulations, it should be a low-hanging fruit to perform a more comparative analysis of meteorology (precipitation, winds, PBL height, etc.), in addition to the present Fig. S2 of temperature comparison. The outcome of this analysis should be also discussed in the context with the recent studies focusing on the air quality changes during the COVID-19.

**Response**:

Thanks for the comments. Considering the important influence of meteorological conditions, we added more analysis on their differences between pre-lockdown and lockdown periods including temperature (T), relative humidity (RH), planetary boundary layer (PBL) height, the average daily precipitation, and wind fields in below Fig. 1 (also added as Fig.S3 in the revised supplement). We also explained the impacts of these meteorological conditions on air quality in the Results and Discussion section.

[Figure]

**Figure 1: Distribution of simulated temperature (T), relative humidity (RH), planetary boundary layer (PBL) height, the average daily precipitation, and wind fields in India before and during the lockdown period. "Case2 - Case1" indicates (Case 2 – Case 1)/Case 1, reported as %.**

**Changes in manuscript:**

**Results and discussion (Lines 199-203 in the revision):** "Variations in near-surface meteorological

factors during lockdown also play an important role in PM$_{2.5}$ changes. As is shown in Fig. S3, lower PM$_{2.5}$ in urban areas during lockdown (Fig. 4) may attribute to the decrease of RH and increase of planetary boundary layer (PBL) height, while the decrease of precipitation and WS allows PM$_{2.5}$ to accumulate in some rural areas (Schnell et al., 2018; Le et al., 2020)."

**Results and discussion (Lines 207-211 in the revision):** "Although significant reductions are found in O$_3$ precursor emissions throughout India during the lockdown, the MDA8 O$_3$ has not shown comparable decreasing trends, which is affected by the meteorological conditions such as an increase of temperature and decrease of RH (Fig. S3). Higher temperature speeds up photochemical processes that produce O$_3$, while higher RH reduces them (Chen et al., 2019; Zhao et al., 2017; Ali et al., 2012)."

**Comments:** There is no doubt that HCHO is closely linked with VOCs, but it is unclear to me how well the HCHO can be used to represent the total VOCs in a quantitative manner. Can the authors show their correlations based on the CMAQ model results?

**Response:**

As one of the most abundant oxygenated VOCs, HCHO is one of the major contributors to total VOCs reactivity (Zhang et al., 2012; Steiner et al., 2008). Therefore, it is used to show the model performance on VOCs due to the lack of VOCs observations. Figure 2 (added as Fig.S4 in the revised supplement) shows scatter plots comparing the simulated average daily HCHO and the total VOCs at all 117×117 grids during the study period. It can be seen from the results that HCHO has a high correlation with VOCs, and R$^2$ reaches 0.93. We added explanations in the manuscript to make it clear.

[Figure]

**Figure 2: Scatter plots comparing the simulated average daily HCHO and the total VOCs at all 117×117 grids from February 21 to April 24, 2020.**

**Changes in manuscript:**

**Results and discussion (Lines 267-269 in the revision):** "We investigated the changes of MDA8 $O_3$ and its major precursors $NO_x$ and HCHO that is one of the major contributors to total VOCs reactivity (Zhang et al., 2012; Steiner et al., 2008) during the lockdown period. Figure S4 shows that HCHO has a strong correlation with total VOCs (R up to 0.93)."

**Comments:** Spaceborne measurements of HCHO and $NO_2$ are available from the latest satellites, such as TROPOMI. There are near-real-time products. The authors may want to explore those data and validate the modeled HCHO and $NO_2$ and see their changes in observations.

**Response**:

Thank you for this good suggestion. To validate the modeled HCHO and $NO_2$, we compared our simulated results with satellite-observed $NO_2$ and HCHO column number density datasets from TROPOMI during pre-lockdown and lockdown periods (Fig. 3, added as Fig. S1 in the revised supplement). As shown in Fig. 3, the predicted regional distribution of $NO_2$ and HCHO column number densities is similar to satellite-observations. Overall, HCHO and $NO_2$ are higher in eastern and northern India than in other regions. And their variation trends from CMAQ and TROPOMI are consistent that $NO_2$ decreases while HCHO increases during the lockdown.

[Figure]

**Figure 3: Comparison of the simulated and satellite-observed NO₂ and HCHO column concentrations before lockdown and during the lockdown in India. The unit is $10^{15}$ molec cm⁻².**

**Changes in manuscript:**

**Methodology (Lines 85-89 in the revision):** "The satellite-observed NO₂ and formaldehyde (HCHO) column number density datasets are from the Sentinel-5 Precursor TROPOspheric Monitoring Instrument (S-5P TROPOMI) (https://scihub.copernicus.eu). Besides, we filter the satellite data under the recommended criteria of QA values greater than 75% for tropospheric NO₂ column number density datasets and 50% for HCHO (Apituley, 2018)."

**Results and discussion (Lines 158-163 in the revision):** "To further validate modeled HCHO and NO₂, we compared our simulated results with satellite-observed data during pre-lockdown and lockdown

periods (Fig. S1). The tropospheric column densities of $NO_2$ and HCHO were calculated by summing their concentrations of 17 vertical layers in the CMAQ model (H. J. Eskes, 2020). The predicted regional distribution of tropospheric column $NO_2$ and HCHO is similar to satellite-observations. Overall, HCHO and $NO_2$ are higher in eastern and northern India than in other regions. And their variation trends from CMAQ and TROPOMI are consistent that $NO_2$ decreases while HCHO increases during the lockdown."

**Reference**

Ali, K., Inamdar, S. R., Beig, G., Ghude, S., and Peshin, S.: Surface ozone scenario at Pune and Delhi during the decade of 1990s, Journal of Earth System Science, 121, 373-383, https://doi.org/10.1007/s12040-012-0170-1, 2012.

Apituley, A., Pedergnana, M., Sneep, M., Pepijn Veefkind, J., Loyola, D., Landgraf, J., Borsdorff, T.: Sentinel-5 Precursor/TROPOMI Level 2 Product User Manual Carbon Monoxide, . Royal Netherlands Meteorological Institute., 2018.

Chen, Z., Zhuang, Y., Xie, X., Chen, D., Cheng, N., Yang, L., and Li, R.: Understanding long-term variations of meteorological influences on ground ozone concentrations in Beijing During 2006–2016, Environmental Pollution, 245, 29-37, https://doi.org/10.1016/j.envpol.2018.10.117, 2019.

S5P Mission Performance Centre Nitrogen Dioxide [L2__NO2___] Readme: https://sentinel.esa.int/documents/247904/3541451/Sentinel-5P-Nitrogen-Dioxide-Level-2-Product-Readme-File, 2020.

Le, T., Wang, Y., Liu, L., Yang, J., Yung, Y. L., Li, G., and Seinfeld, J. H.: Unexpected air pollution with marked emission reductions during the COVID-19 outbreak in China, Science, 369, 702, https://doi.org/10.1126/science.abb7431, 2020.

Schnell, J. L., Naik, V., Horowitz, L. W., Paulot, F., Mao, J., Ginoux, P., Zhao, M., and Ram, K.: Exploring the relationship between surface $PM_{2.5}$ and meteorology in Northern India, Atmospheric Chemistry and Physics, 18, 10157-10175, https://doi.org/10.5194/acp-18-10157-2018, 2018.

Steiner, A. L., Cohen, R. C., Harley, R. A., Tonse, S., Millet, D. B., Schade, G. W., and Goldstein, A. H.: VOC reactivity in central California: comparing an air quality model to ground-based measurements, Atmos. Chem. Phys., 8, 351-368, https://doi.org/10.5194/acp-8-351-2008, 2008.

Zhang, Q., Shao, M., Li, Y., Lu, S. H., Yuan, B., and Chen, W. T.: Increase of ambient formaldehyde in Beijing and its implication for VOC reactivity, Chinese Chemical Letters, 23, 1059-1062,

https://doi.org/10.1016/j.cclet.2012.06.015, 2012.

Zhao, B., Wu, W., Wang, S., Xing, J., Chang, X., Liou, K.-N., Jiang, J. H., Gu, Y., Jang, C., Fu, J. S., Zhu, Y., Wang, J., Lin, Y., and Hao, J.: A modeling study of the nonlinear response of fine particles to air pollutant emissions in the Beijing–Tianjin–Hebei region, Atmospheric Chemistry and Physics, 17, 12031-12050, https://doi.org/10.5194/acp-17-12031-2017, 2017.

---

## Author Comment (AC2) · 25 Jan 2021

Dear Dr. Sharma, We appreciate your comments to help improve the manuscript. We tried our best to address your comments and detailed responses and related changes are shown below. Our response is in blue and the modifications in the manuscript are in red. Besides, please note the supplementary PDF file in the reply.

Comments: The paper presents the percentage reduction in concentrations of air pollutants (PM2.5, O3, CO and NO2 etc.,) in India during lockdown period using ground observations (at Delhi, Lucknow, Kolkata, Ahmedabad, Mumbai, Hyderabad, Bangalore and Chennai during 21 February to 24 April 2020) and air quality models (WRF and CMAQ). The paper also described the percent reduction in secondary inorganic

aerosols/species of PM2.5 over the region. A quick search on the web shows numerous studies on these lines published in the various reputed journals over Delhi and other locations of India. Some of them are cited in the present papers. A comprehensive study on the air quality of India during and before lockdown period has been carried by Kumar et al. (2020) in Sustainable Cities and Society (2020): 102382 and they have also compared most of the recent studies. The only advantage of this manuscript is that the authors have used the 7 stations ground based observation datasets (PM2.5, O3, CO and NO2 etc.,), validated the WRF and CMAQ models and extrapolated the pollutants for the country. This advantage gives the merit for publication of the paper in journal. However, the following points may also be considered before publication in ACP. Response: We appreciate the positive comments and helpful suggestions from Dr. Sharma on this paper. Below is the response to each specific comment.

Comments: Authors are suggested to search all the recent articles published on these line in various journal over Indian and other region of the globe and make a summary Table and include in Introduction section. Also highlights how the present study is different from the previous studies as well new scientific information are going to be add in existing knowledge. Response: Thanks for the comments. The summary Table including recent studies about COVID-19 has been summarized by Kumar et al. (2020) (Table 1 in their paper) comprehensively before, so we didn't duplicate the table in our manuscript. In the revision, more discussions about previous studies are added to reinforce the comparison with previous studies in the Introduction section. Changes in manuscript: Introduction (Lines 52-53 in the revision): "Therefore, it is significant to understand the mechanisms involving in air pollution formation before and after dramatic emission changes comprehensively, in addition to the comparison of air pollution levels." Introduction (Lines 55-56 in the revision): "Srivastava et al. (2020) reported the concentrations of primary air pollutants are drastically lowed as a result of emission reduction." Introduction (Lines 57-59 in the revision): "These studies pointed out that the air quality was improved during the lockdown period compared with the period before lockdown and depends on the duration of the lockdown (Kumar et al., 2020; Mor et al.,
2021)." Introduction (Lines 59-64 in the revision): "Besides, compared with the same period in previous years, Gautam (2020) claimed that aerosol concentration levels are at their lowest in the last 20 years during lockdown based on satellite data. Selvam et al. (2020) stated that Air Quality Index (AQI) was improved by 58% in Gujarat state of western India during lockdown (March 24, 2020 – April 20, 2020) compared to 2019. Kabiraj and Gavli (2020) concluded that the mean concentration of PM2.5 decreased by 42.25% from January to May in 2020 compared with 2019. Similarly, Das et al. (2020) also showed that great reductions of PM2.5 were found across cities in the Indo-Gangetic Plain (IGP) compared with 2018 and 2019."

Comments: A number of publications are published recently on temporary reduction of concentrations of pollutants over India and other country during lockdown period. Hence, the present paper should be scientifically different and should add new information for scientific community. Response: We appreciate the comments. We comprehensively evaluate the impact of the nationwide lockdown on air quality in India, which also provides reliable recommendations for the improvement of emission reduction policies. First, we determined the response of air quality in India under the synergetic impacts from the meteorological conditions and anthropogenic emissions during the pre-lockdown and lockdown periods. For instance, we directly quantified the change in air quality during lockdown due to the reduced anthropogenic emissions through differences between Case 1 (without emission reductions) and Case 2 (with emission reductions) during the lockdown. This casts lights on the policy implementation in India, which may help to mitigate air pollution in the future. Second, we are the first study that explored the impacts of COVID-19 lockdown on Indian air quality on a regional scale. It allows us to figure out the changes of primary and even secondary pollutants during two periods (pre-lockdown and lockdown) and illustrate their differences in urban and rural areas. This could be a great help to formulate the city-level control policy in India. Third, in atmospheric chemistry, we developed a better understanding of the secondary pollutants formations by investigating their non-linear responses to the precursors' changes during the lockdown. In particular, the sensitivity of PM2.5 secondary

components (Fig. 6 in the revised manuscript) and the change of spatial distributions of O3 production sensitivity (Fig. S5 in the revised supplement) due to emission changes during the lockdown give us a more in-depth discussion on secondary pollutants. In the revised manuscript, we added such information to the Introduction to make it clear. Changes in manuscript: Introduction (Lines 64-68 in the revision): "However, the role of meteorological conditions and chemical reactions involving changes in air quality is not clear from these observation-based studies, which only showed the phenomenon of concentration reduction and switch of major primary pollutants mainly in urban cities. Further, the number of monitoring stations in the country is way below the guidelines by the governing bodies and not uniformly distributed, which results in observation data limitations in India (Sahu et al., 2020)." Introduction (Lines 69-74 in the revision): "In this study, the Community Multi-Scale Air Quality (CMAQ) model was used to investigate changes in air pollutants during the pre-lockdown (from February 21, 2020 to March 23, 2020) and lockdown (from March 24, 2020 to April 24, 2020) periods throughout Indian region. We explored the synergetic impacts from the meteorological conditions and anthropogenic emissions during the pre-lockdown and lockdown periods. Besides, we directly quantified the change in air quality during the lockdown due to the reduced anthropogenic emissions by comparing the differences between Case 1 (without emission reductions) and Case 2 (with emission reductions)." Conclusion (Lines 308-309 in the revision): "However, more stringent mitigation measures are needed to achieve effective control of air pollution from secondary air pollutants and their components, particularly in rural areas."

Comments: Conclusion sections may be improved with avoiding the repetitive information from the abstract. Response: Thanks for the comments. We removed duplicate information from the Conclusion section and added policy recommendations. Changes in manuscript: Conclusion (Lines 304-305 in the revision): "The drastic decline in PM2.5 and its major components during the lockdown period in Case 2 compared with Case 1 shows the positive impacts of emission control measures, especially for SIA." Conclusion (Lines 308-309 in the revision): "However, more stringent mitigation

measures are needed to achieve effective control of air pollution from secondary air pollutants and their components, particularly in rural areas."

Please also note the supplement to this comment:
https://acp.copernicus.org/preprints/acp-2020-903/acp-2020-903-AC2-supplement.pdf

[Figure]

**Supplement:**

**Responses to interactive comments**

**Journal: Atmospheric Chemistry and Physics**

**Manuscript ID: acp-2020-903**

**Title: "Impact of reduced anthropogenic emissions during COVID-19 on air quality in India"**

Dear Dr. Sharma,

We appreciate your comments to help improve the manuscript. We tried our best to address your comments and detailed responses and related changes are shown below. Our response is in blue and the modifications in the manuscript are in red.

**Comments:** The paper presents the percentage reduction in concentrations of air pollutants ($PM_{2.5}$, $O_3$, CO and $NO_2$ etc.,) in India during lockdown period using ground observations (at Delhi, Lucknow, Kolkata, Ahmedabad, Mumbai, Hyderabad, Bangalore and Chennai during 21 February to 24 April 2020) and air quality models (WRF and CMAQ). The paper also described the percent reduction in secondary inorganic aerosols/species of $PM_{2.5}$ over the region. A quick search on the web shows numerous studies on these lines published in the various reputed journals over Delhi and other locations of India. Some of them are cited in the present papers. A comprehensive study on the air quality of India during and before lockdown period has been carried by Kumar et al. (2020) in Sustainable Cities and Society (2020): 102382 and they have also compared most of the recent studies.

The only advantage of this manuscript is that the authors have used the 7 stations ground based observation datasets ($PM_{2.5}$, $O_3$, CO and $NO_2$ etc.,), validated the WRF and CMAQ models and extrapolated the pollutants for the country. This advantage gives the merit for publication of the paper in journal. However, the following points may also be considered before publication in ACP.

**Response:**

We appreciate the positive comments and helpful suggestions from Dr. Sharma on this paper. Below is the response to each specific comment.

**Comments:** Authors are suggested to search all the recent articles published on these line in various journal over Indian and other region of the globe and make a summary Table and include in Introduction

section. Also highlights how the present study is different from the previous studies as well new scientific information are going to be add in existing knowledge.

**Response**:

Thanks for the comments. The summary Table including recent studies about COVID-19 has been summarized by Kumar et al. (2020) (Table 1 in their paper) comprehensively before, so we didn't duplicate the table in our manuscript. In the revision, more discussions about previous studies are added to reinforce the comparison with previous studies in the Introduction section.

**Changes in manuscript:**

**Introduction (Lines 52-53 in the revision):** "Therefore, it is significant to understand the mechanisms involving in air pollution formation before and after dramatic emission changes comprehensively, in addition to the comparison of air pollution levels."

**Introduction (Lines 55-56 in the revision):** "Srivastava et al. (2020) reported the concentrations of primary air pollutants are drastically lowed as a result of emission reduction."

**Introduction (Lines 57-59 in the revision):** "These studies pointed out that the air quality was improved during the lockdown period compared with the period before lockdown and depends on the duration of the lockdown (Kumar et al., 2020; Mor et al., 2021)."

**Introduction (Lines 59-64 in the revision):** "Besides, compared with the same period in previous years, Gautam (2020) claimed that aerosol concentration levels are at their lowest in the last 20 years during lockdown based on satellite data. Selvam et al. (2020) stated that Air Quality Index (AQI) was improved by 58% in Gujarat state of western India during lockdown (March 24, 2020 – April 20, 2020) compared to 2019. Kabiraj and Gavli (2020) concluded that the mean concentration of $PM_{2.5}$ decreased by 42.25% from January to May in 2020 compared with 2019. Similarly, Das et al. (2020) also showed that great reductions of $PM_{2.5}$ were found across cities in the Indo-Gangetic Plain (IGP) compared with 2018 and 2019."

**Comments:** A number of publications are published recently on temporary reduction of concentrations of pollutants over India and other country during lockdown period. Hence, the present paper should be scientifically different and should add new information for scientific community.

**Response**:

We appreciate the comments. We comprehensively evaluate the impact of the nationwide lockdown on air quality in India, which also provides reliable recommendations for the improvement of emission reduction policies.

First, we determined the response of air quality in India under the synergetic impacts from the meteorological conditions and anthropogenic emissions during the pre-lockdown and lockdown periods. For instance, we directly quantified the change in air quality during lockdown due to the reduced anthropogenic emissions through differences between Case 1 (without emission reductions) and Case 2 (with emission reductions) during the lockdown. This casts lights on the policy implementation in India, which may help to mitigate air pollution in the future.

Second, we are the first study that explored the impacts of COVID-19 lockdown on Indian air quality on a regional scale. It allows us to figure out the changes of primary and even secondary pollutants during two periods (pre-lockdown and lockdown) and illustrate their differences in urban and rural areas. This could be a great help to formulate the city-level control policy in India.

Third, in atmospheric chemistry, we developed a better understanding of the secondary pollutants formations by investigating their non-linear responses to the precursors' changes during the lockdown. In particular, the sensitivity of $PM_{2.5}$ secondary components (Fig. 6 in the revised manuscript) and the change of spatial distributions of $O_3$ production sensitivity (Fig. S5 in the revised supplement) due to emission changes during the lockdown give us a more in-depth discussion on secondary pollutants.

In the revised manuscript, we added such information to the Introduction to make it clear.

**Changes in manuscript:**

**Introduction (Lines 64-68 in the revision):** "However, the role of meteorological conditions and chemical reactions involving changes in air quality is not clear from these observation-based studies, which only showed the phenomenon of concentration reduction and switch of major primary pollutants mainly in urban cities. Further, the number of monitoring stations in the country is way below the guidelines by the governing bodies and not uniformly distributed, which results in observation data limitations in India (Sahu et al., 2020)."

**Introduction (Lines 69-74 in the revision):** "In this study, the Community Multi-Scale Air Quality (CMAQ) model was used to investigate changes in air pollutants during the pre-lockdown (from February 21, 2020 to March 23, 2020) and lockdown (from March 24, 2020 to April 24, 2020) periods throughout Indian region. We explored the synergetic impacts from the meteorological conditions and

anthropogenic emissions during the pre-lockdown and lockdown periods. Besides, we directly quantified the change in air quality during the lockdown due to the reduced anthropogenic emissions by comparing the differences between Case 1 (without emission reductions) and Case 2 (with emission reductions)."

**Conclusion (Lines 308-309 in the revision):** "However, more stringent mitigation measures are needed to achieve effective control of air pollution from secondary air pollutants and their components, particularly in rural areas."

**Comments:** Conclusion sections may be improved with avoiding the repetitive information from the abstract.

**Response**: Thanks for the comments. We removed duplicate information from the Conclusion section and added policy recommendations.

**Changes in manuscript:**

**Conclusion (Lines 304-305 in the revision):** "The drastic decline in $PM_{2.5}$ and its major components during the lockdown period in Case 2 compared with Case 1 shows the positive impacts of emission control measures, especially for SIA."

**Conclusion (Lines 308-309 in the revision):** "However, more stringent mitigation measures are needed to achieve effective control of air pollution from secondary air pollutants and their components, particularly in rural areas."

**Reference**

Das, M., Das, A., Sarkar, R., Saha, S., and Mandal, A.: Examining the impact of lockdown (due to COVID-19) on ambient aerosols ($PM_{2.5}$): A study on Indo-Gangetic Plain (IGP) Cities, India, Stochastic Environmental Research and Risk Assessment, https://doi.org/10.1007/s00477-020-01905-x, 2020.

Gautam, S.: The Influence of COVID-19 on Air Quality in India: A Boon or Inutile, Bulletin of Environmental Contamination and Toxicology, 104, 724-726, https://doi.org/10.1007/s00128-020-02877-y, 2020.

Kabiraj, S., and Gavli, N. V.: Impact of SARS-CoV-2 Pandemic Lockdown on Air Quality Using Satellite Imagery with Ground Station Monitoring Data in Most Polluted City Kolkata, India, Aerosol Science and Engineering, 4, 320-330, https://doi.org/10.1007/s41810-020-00077-z, 2020.

Kumar, P., Hama, S., Omidvarborna, H., Sharma, A., Sahani, J., Abhijith, K. V., Debele, S. E., Zavala-

Reyes, J. C., Barwise, Y., and Tiwari, A.: Temporary reduction in fine particulate matter due to 'anthropogenic emissions switch-off' during COVID-19 lockdown in Indian cities, Sustainable Cities and Society, 62, 102382, https://doi.org/10.1016/j.scs.2020.102382, 2020.

Mor, S., Kumar, S., Singh, T., Dogra, S., Pandey, V., and Ravindra, K.: Impact of COVID-19 lockdown on air quality in Chandigarh, India: Understanding the emission sources during controlled anthropogenic activities, Chemosphere, 263, 127978, https://doi.org/10.1016/j.chemosphere.2020.127978, 2021.

Sahu, S. K., Sharma, S., Zhang, H., Chejarla, V., Guo, H., Hu, J., Ying, Q., Xing, J., and Kota, S. H.: Estimating ground level $PM_{2.5}$ concentrations and associated health risk in India using satellite based AOD and WRF predicted meteorological parameters, Chemosphere, 255, https://doi.org/10.1016/j.chemosphere.2020.126969, 2020.

Selvam, S., Muthukumar, P., Venkatramanan, S., Roy, P. D., Manikanda Bharath, K., and Jesuraja, K.: SARS-CoV-2 pandemic lockdown: Effects on air quality in the industrialized Gujarat state of India, Science of The Total Environment, 737, 140391, https://doi.org/10.1016/j.scitotenv.2020.140391, 2020.

Srivastava, S., Kumar, A., Bauddh, K., Gautam, A. S., and Kumar, S.: 21-Day Lockdown in India Dramatically Reduced Air Pollution Indices in Lucknow and New Delhi, India, Bulletin of Environmental Contamination and Toxicology, 105, 9-17, https://doi.org/10.1007/s00128-020-02895-w, 2020.

---

## Author Comment (AC3) · 25 Jan 2021

Dear Referee #3, We appreciate your comments to help improve the manuscript. We tried our best to address your comments and detailed responses and related changes are shown below. Our response is in blue and the modifications in the manuscript are in red. Besides, please note the supplementary PDF file in the reply.

Comments: The paper by Zhang et al. entitled "Impact of reduced anthropogenic emissions during COVID-19 on air quality in India" is on a very relevant and interesting topic which is to use the covid lockdown emission reductions for assessing impacts on air quality over India. Unfortunately, the analyses and interpretation are weak in several places (listed below). There are hardly any new trustworthy insights

from this modelling study which have not been reported already by the authors in previous works on the same topic published recently (see Sharma, S., Zhang, M., Anshika, Gao, J., Zhang, H., and Kota, S. H.: Effect of restricted emissions during COVID-19 on air quality in India, Science of The Total Environment, 728, 138878, https://doi.org/10.1016/j.scitotenv.2020.138878, 2020). Response: We appreciate the comments. However, we don't agree that there are hardly any new trustworthy insights. We comprehensively evaluate the impact of the nationwide lockdown on air quality in India, which also provides reliable recommendations for the improvement of emission reduction policies. First, we determined the response of air quality in India under the synergetic impacts from the meteorological conditions and anthropogenic emissions during the pre-lockdown and lockdown periods. For instance, we directly quantified the change in air quality during lockdown due to the reduced anthropogenic emissions through differences between Case 1 (without emission reductions) and Case 2 (with emission reductions) during the lockdown. This casts lights on the policy implementation in India, which may help to mitigate air pollution in the future. Second, we are the first study that explored the impacts of COVID-19 lockdown on Indian air quality on a regional scale. It allows us to figure out the changes of primary and even secondary pollutants during two periods (pre-lockdown and lockdown) and illustrate their differences in urban and rural areas. This could be a great help to formulate the city-level control policy in India. Third, in atmospheric chemistry, we developed a better understanding of the secondary pollutants formations by investigating their non-linear responses to the precursors' changes during the lockdown. In particular, the sensitivity of PM2.5 secondary components (Fig. 6 in the revised manuscript) and the change of spatial distributions of O3 production sensitivity (Fig. S5 in the revised supplement) due to emission changes during the lockdown give us a more in-depth discussion on secondary pollutants. In the revised manuscript, we added such information to the Introduction to make it clear. Changes in manuscript: Introduction (Lines 64-68 in the revision): "However, the role of meteorological conditions and chemical reactions involving changes in air quality is not clear from these observation-based studies,

which only showed the phenomenon of concentration reduction and switch of major primary pollutants mainly in urban cities. Further, the number of monitoring stations in the country is way below the guidelines by the governing bodies and not uniformly distributed, which results in observation data limitations in India (Sahu et al., 2020)." Introduction (Lines 69-74 in the revision): "In this study, the Community Multi-Scale Air Quality (CMAQ) model was used to investigate changes in air pollutants during the pre-lockdown (from February 21, 2020 to March 23, 2020) and lockdown (from March 24, 2020 to April 24, 2020) periods throughout Indian region. We explored the synergetic impacts from the meteorological conditions and anthropogenic emissions during the pre-lockdown and lockdown periods. Besides, we directly quantified the change in air quality during the lockdown due to the reduced anthropogenic emissions by comparing the differences between Case 1 (without emission reductions) and Case 2 (with emission reductions)." Conclusion (Lines 309-310 in the revision): "However, more stringent mitigation measures are needed to achieve effective control of air pollution from secondary air pollutants and their components, particularly in rural areas."

Comments: Instead there are even discrepancies from the earlier work based on interpretation of what appears to be the same measured dataset. While in the previous work (Sharma et al., 2020) it was reported that there was a 17% increase in ozone during COVID, in the present work it has been reported that a significant decrease in surface ozone (MDA8 values) occurred, without even clarifying what changed between the two studies except for additional modelling analyses in this study. Response: We are sorry for being not clear enough. As described in the previous response, these two studies are different in many aspects and we are not aiming to show the same results. First, the duration of lockdown considered in this study (from March 24 to April 24, 2020) is different from Sharma et al. (from March 15th to April 14, 2020). Second, the variations of MDA8 O3 in this study included all India within the 36-km domain ($117\times117$ grids) (Fig. 1 in the manuscript), while Sharma et al. only focused on urban measurements at 22 cities. Third, this study excluded the influence of meteorology by comparing Case 1 (without emission reductions) and Case 2 (with emission reductions) during the lockdown, while Sharma et al. concluded the increase in O3 by comparing 2020 and the previous three years. Moreover, the results of our study are consistent with Sharma et al. in the urban areas. In these areas that were under VOC-limited conditions, both studies concluded that O3 increased during the lockdown. In our study, the increase in MDA8 O3 was up to 21%, which was close to Shame et al. In summary, different research methods and study periods result in different O3 changing ratios, and it is not able to conclude that as discrepancies. Changes in manuscript: Since the differences have been added in the previous comment, no special changes were made for this point.

Comments: There are several major issues with the present submission which need to be addressed/clarified for meriting further publication in ACP. Response: We thank the reviewers for the detailed comments below and made necessary changes to the manuscript.

Comments: Validation of the model used in this work has not been done/described adequately: Authors use only 2 m level measurements of temperature and meteorological data and chemical data from 5 monitoring stations operated by the regulatory agency of India located within cities to compare their modelled output. Response: We collected all available observations to validate our WRF and CMAQ models. Since the observations are limited in India, it is important to conduct simulation studies like this one to improve our understanding and help design control strategies. Changes in manuscript: No changes were made for this point.

Comments: Measured chemical data: The authors present only daily averaged data in the plots (Figures 2 and 3). This would be fine but I could find no details of the original high resolution primary data (presumably available at temporal resolution of few minutes from the analyzers in the monitoring stations) to build confidence in the reader about the trustworthiness of the primary data and its quality assurance. If they could provide such high resolution data for the five stations (even for few days in both periods) for ozone, NO, NO2, PM2.5 etc.. with gaps in measurements if any (after all

there was a lockdown so maintenance could be difficult), and the calibration data of any of the analyzers, it would go a long way in instilling confidence in the highly averaged data. The reviewer looked up their previous study Sharma et al 2020 which has been cited for detailed description of the primary data and found that this reference did not contain these details and somewhat remarkably the Sharma et al. 2020 paper reported data until April 14th 2020 in that work, was submitted on April 16th, 2020 and accepted on April 19th, 2020. While this does not necessarily suggest that due diligence was not taken as given the nature of topic urgency to publish would have been a factor, the rapid turn-around time and lack of experimental details in the peer reviewed reference cited and which forms the basis of the daily averages does leave room for concern. So the authors should provide the original primary data as a time series for these 5 monitoring stations in the revised supplement along with details of calibration experiments and data quality control followed to allay such potential concerns about the primary measured dataset. Response: Although higher resolution from some sampling equipment is available, monitoring agencies worldwide only report hourly data. The data from the CPCB database can be downloaded at https://app.cpcbccr.com/ccr/#/caaqm-dashboard-all/caaqm-landing/data and we are not able to post the raw data here due to the CPCB's data access statement. Also, the temporal-resolution of the CMAQ model is hourly. Besides, daily average PM2.5 is commonly used in previous studies due to its practical significance and data availability. For O3, the maximum daily 8-hour concentration (MDA8 O3) calculated as running the maximum continuous 8-hour data is used to represent pollution level (Lei et al., 2019). Thus, we used the daily average PM2.5 and MDA8 O3 specifically in Figures 2 and 3 in the manuscript. We are sorry that the validity of the observation data is not clear. The observation data from CPCB has been through quality assurance or quality control programs by establishing strict procedures for sampling, analysis, and calibration before publication (Gurjar et al., 2016). Thus, studies usually use the data directly. In this study, we made additional checks to screen out outliers. For example, a cut-off value of 40 ppb was applied to hourly O3 based on EPA's recommendations (EPA, 2005). For PM2.5, abnormally high

values of greater than 300 $\mu$g m-3 were excluded. These explanations were added in different parts of the revised manuscript. Changes in manuscript: Methodology (Lines 82-83 in the revision): "The CPCB database provides data quality assurance (QA) or quality control (QC) programs by establishing strict procedures for sampling, analysis, and calibration (Gurjar et al., 2016)." Results and discussion (Lines 150-153 in the revision): "For PM2.5, the averaged mean fractional bias (MFB) (-0.48) and mean fractional error (MFE) (0.61) values met the criteria limits of $\pm$0.6 and 0.75 claimed by the EPA (2007b) in all the five urban cites after excluding some abnormally high values of greater than 300 $\mu$g m-3. For O3, a cut-off value of 40 ppb is applied, which is based on EPA's recommendations (EPA, 2005)."

Comments: Also they should discuss whether data from 5 cities are adequate to make inferences about all of India with same degree of confidence which spans vast rural and countryside regions? It might be advisable to better focus on the 5 cities alone for which they have the data and even there they should acknowledge how data from one monitoring station may be limited for representing air quality of the entire city. In fact, a combination of monitoring station data and satellite data (agreed also can have issues but better than nothing) would be better. Response: Thanks for the comments. First of all, we agree that 5 cities were maybe inadequate to evaluate the model for the whole of India. However, it is an endless effort and impractical to monitor the whole country. Focusing only on areas with observations is a safe idea, but not a promising one. This actually is the advantage of modelling work and the merit of this study. Besides, when we validate our simulated results in urban areas, we have the confidence to investigate rural and countryside regions. The modelling work is not only to reproduce what has been observed but more importantly to investigate what is not been observed after enough validation. If the monitoring station can represent the entire city is another good question for observation experts. In this study, the observation data we compare with the predicted values are an average of multiple sites in each city (Table 1 in the attached PDF file), which can represent the PM2.5 and O3 levels of the entire city well. We also added more observations to make the results more

representative in the revision (Fig. 1, Fig. 2). The model performed well at simulating O3, PM2.5, CO, and NO2 in these major cities in India (Table 2 in the attached PDF file). Thanks to the suggestion of using satellite, we compared model performance with satellite observations (TROPOMI) for HCHO and NO2 (Fig. 3). The corresponding explanatory statements were added in the Results and discussion section. Changes in manuscript: Results and discussion (Lines 150-153 in the revision): "For PM2.5, the averaged mean fractional bias (MFB) (-0.48) and mean fractional error (MFE) (0.61) values met the criteria limits of $\pm0.6$ and 0.75 claimed by the EPA (2007b) in all the five urban cites after excluding some abnormally high values of greater than 300 $\mu$g m-3. For O3, a cut-off value of 40 ppb is applied, which is based on EPA's recommendations (EPA, 2005)." Results and discussion(Lines 154-156 in the revision): "And averaged MFB (-0.05) and MFE (0.25) values of O3 also satisfy the benchmarks of $\pm0.15$ and 0.30 set by the EPA (2005) in most of these cities with Chennai and Hyderabad exceeding the limits slightly." Results and discussion (Lines 168-172 in the revision): "Overall, sharp decreases are found in the observed PM2.5 in all these cities, and the averaged PM2.5 level drops from 43.18 $\mu$g m-3 to 27.62 $\mu$g m-3. The mean observed PM2.5 concentrations during lockdown are 42.47 $\mu$g m-3 (Delhi), 24.53 $\mu$g m-3 (Mumbai), 15.73 $\mu$g m-3 (Chennai), 31.29 $\mu$g m-3 (Hyderabad), 24.08 $\mu$g m-3 (Bengaluru), which are reduced by 41%, 40%, 42%, 10%, and 43% respectively compared with that of the pre-lockdown period. Besides, the observed peak values of PM2.5 in each city also decrease appreciably (up to 57%) during the lockdown period." Results and discussion (Lines 177-178 in the revision): "The observed average MDA8 O3 during lockdown is higher than that of pre-lockdown in Delhi (2%), Hyderabad (12%), and Bengaluru (2%)." Results and discussion (Lines 182-183 in the revision): "In contrast, the observed average MDA8 O3 during lockdown is reduced compared with the pre-lockdown period in both Mumbai (-35%) and Chennai (-13%)." Methodology (Lines 85-89 in the revision): "The satellite-observed NO2 and formaldehyde (HCHO) column number density datasets are from the Sentinel-5 Precursor TROPOspheric Monitoring Instrument (S-5P TROPOMI) (https://scihub.copernicus.eu). Besides, we filter the satellite data under

the recommended criteria of QA values greater than 75% for tropospheric NO2 column number density datasets and 50% for HCHO (Apituley, 2018)." Results and discussion (Lines 158-163 in the revision): "To further validate modeled HCHO and NO2, we compared our simulated results with satellite-observed data during pre-lockdown and lockdown periods (Fig. S1). The tropospheric column densities of NO2 and HCHO were calculated by summing their concentrations of 17 vertical layers in the CMAQ model (H. J. Eskes, 2020). The predicted regional distribution of tropospheric column NO2 and HCHO is similar to satellite-observations. Overall, HCHO and NO2 are higher in eastern and northern India than in other regions. And their variation trends from CMAQ and TROPOMI are consistent that NO2 decreases while HCHO increases during the lockdown."

Comments: Validation of model using 2 m level measured meteorological data: For the kind of modelling investigation the authors are making namely, effect of emission changes on concentrations of pollutants use of only the 2 m level observations without comparison with satellite data, sonde data, mixing layer height data (see ERA5 products) seems to be a major shortcoming. Note that the changes in ventilation coefficient before and during lockdown and the changing season (Spring to Summer) can alone have big impacts on the concentrations. Response: Thanks for the referee's comments. The weather research and forecasting (WRF) ARW regional model is a high‐resolution meteorology model that is widely used in Indian studies (Gevorgyan, 2018; Srinivas et al., 2013; Chawla et al., 2018; Ashrit and Mohandas, 2010). The capability of the WRF model has been validated for providing reliable meteorological inputs to air quality models even under extreme weather events (Zhang et al., 2020; Stella and Agnihotri, 2015; Pattanaik and Rama Rao, 2009; Rajeevan et al., 2010). In the manuscript, we validated the model performance of WRF using observation from the National Climate Data Center (NCDC). Although only the near-surface meteorological factors are considered, our prediction also shows good performance with the comparison of satellite data (NO2 and HCHO) (Fig. 3). As suggested by the referee, the reanalysis-based ERA5 product produces global atmospheric quantities at 31-km horizontal resolution combining model simulations and observations. However, there are uncertainties such as its high root-mean-square error values in the tropical and subtropical climate zone so that the reliability and applicability of the ERA5 dataset still need to be explored in India with a tropical monsoon climate (Jiang et al., 2020; Kolluru et al., 2020). In follow-up studies, we will consider its application to further validate the WRF model. As for the impacts of ventilation coefficient and changing season, we added more analysis of the difference in meteorological conditions between pre-lockdown and lockdown periods including temperature (T), relative humidity (RH), planetary boundary layer (PBL) height, the average daily precipitation, and wind fields in Fig. 4. The explanations about the impacts of these meteorological conditions on Indian air quality are also added in the Results and discussion section. Changes in manuscript: Results and discussion (Lines 199-203 in the revision): "Variations in near-surface meteorological factors during lockdown also play an important role in PM2.5 changes. As is shown in Fig. S3, lower PM2.5 in urban areas during lockdown (Fig. 4) may attribute to the decrease of RH and increase of planetary boundary layer (PBL) height, while the decrease of precipitation and WS allows PM2.5 to accumulate in some rural areas (Schnell et al., 2018; Le et al., 2020)." Results and discussion (Lines 207-211 in the revision): "Although significant reductions are found in O3 precursor emissions throughout India during the lockdown, the MDA8 O3 has not shown comparable decreasing trends, which is affected by the meteorological conditions such as an increase of temperature and decrease of RH (Fig. S3). Higher temperature speeds up photochemical processes that produce O3, while higher RH reduces them (Chen et al., 2019; Zhao et al., 2017; Ali et al., 2012)." Conclusion (Lines 302-303 in the revision): "It can be concluded that meteorological conditions play an important role in those increases according to the comparison between pre-lockdown (Case 1), and lockdown (Case 2)."

Comments: Changes in atmospheric chemistry of primary pollutant removal and formation of secondary pollutants: Currently the study tends to attribute all the observed concentration changes in pollutants primarily to the emission reductions. However it

has been documented elegantly in the following paper: Kroll, J.H., Heald, C.L., Cappa, C.D. et al. The complex chemical effects of COVID-19 shutdowns on air quality. Nat. Chem. 12, 777–779 (2020). https://doi.org/10.1038/s41557-020-0535-z, that several other processes play a big role. The purpose of using a model should be that these effects can be teased out through sensitivity experiments but unfortunately this has not been addressed in current version of the manuscript. For example the authors note that the temperature increased during the lockdown period. A key question is what effect the temperature change and the reduced emission of VOCs (no VOC measurements have been provided at all), NOx and CO would have on the removal rates of primary pollutants and formation of secondary pollutants. Further have authors identified days when it rained in both pre covid lockdown and during lockdown periods which would cause strong biases for the comparisons. Response: We showed not only the impact of emission reductions but also that the meteorology conditions and explained that the specific chemical reactions can affect the change of pollutant concentrations. We also added more comprehensive discussions about the effects of chemical processes in the manuscript (see the Results and discussion section). We utilized the model to explore the comprehensive effects of meteorology including the increased temperature on the change of primary and secondary pollutants by comparing pre-lockdown (Case 1) and lockdown (Case 2) rather than focusing on a single variable. We also investigated the nonlinear relationship between emissions and atmospheric composition. For example, Fig. 8 in the manuscript also showed the relative changes in concentrations of HCHO and NOx and emissions of HCHO and NOx in eight major cities of India in Case 2 to Case 1 and proved that the concentration of NOx is appreciably highly sensitive to a primary NOx emission reduction compared with HCHO. In specific, though we can't obtain a reliable VOC observation dataset, the high correlation between HCHO and the total VOCs is shown in Fig. 5 below, and HCHO has been further validated by satellite-observations (Fig. 3). We also added discussion on the effects of variations in precipitation during the lockdown on PM2.5 and MDA8 O3 changes in the Results and discussion results. In addition to the regional change of precipitation shown in Fig.

4, we also added the daily average precipitation figures in these 5 major cities from February 21 to April 24, 2020 (Fig. 6 in the response). On average, the precipitation in India is relatively low from the pre-lockdown to lockdown periods (lower than 1 mm for each city). Although the few rainy days (such as March 5, 2020) may promote PM2.5 removal, generally, it has little impact on the comparison of overall air quality before and during the lockdown. Changes in manuscript: Results and discussion (Lines 199-203 in the revision): "Variations in near-surface meteorological factors during lockdown also play an important role in PM2.5 changes. As is shown in Fig. S3, lower PM2.5 in urban areas during lockdown (Fig. 4) may attribute to the decrease of RH and increase of planetary boundary layer (PBL) height, while the decrease of precipitation and WS allows PM2.5 to accumulate in some rural areas (Schnell et al., 2018; Le et al., 2020)." Results and discussion (Lines 241-242 in the revision): "Besides, the reduction of NOx may lead to an increase of SOA offsetting some of the influence caused by the reduction in VOC emissions (Kroll et al., 2020)."

Comments: Existing inadequacies in VOC emission inventories and modelled ozone simulations over India: While the authors are using pre lockdown and lockdown periods for comparison, it is a fact that of all emission inventories, VOC emissions are the most poorly constrained due to the absence of in-situ VOC data over many regions in India. A generic problem also seen is the tendency for overestimation of ozone by models over the Indian region. This suggests that the basic reactant mixture and chemistry are still inadequate for modelling ozone and secondary pollutant formation accurately over India. So how can one be sure that the changed chemical mixture between pre-lockdown and during lockdown are not skewed by these gaps in our basic understanding? While it would be unfair to hold the authors to solve all these issues, one does expect that the limitations and existing issues are duly acknowledged in the work instead of making highly speculative and prescriptive measures for air quality mitigation based on such modelling results. Response: Thanks for pointing out this. We also acknowledge that there are still many limitations and deficiencies in the VOC emission inventories, and the simulation of O3 is usually over-estimated (Kota et al., 2018; Hu et

al., 2015). Much more needs to be done regarding emissions and chemical reactions to better simulate O3 in India. But there is no denying that the simulation results of this model are acceptable compared with the standards recommended by EPA (EPA, 2005, 2007a). In the future, we will continuously apply new information to improve our modelling results once they are available. Changes in manuscript: Results and discussion (Lines 163-164 in the revision): "We also acknowledge that the uncertainty of emission inventory and chemical mechanism in the modelling may affect the simulated results (Dominutti et al., 2020; Kitayama et al., 2019)." Comments: Use of formaldehye for constraining VOC emissions where a large number of more reactive primary VOC emissions occur should also be discussed and clarified. Trusting the formaldehye from the model in absence of in-situ formaldehye measurements to compare with or even satellite or columnar measurements which have been reported from India is recommended. Response: As one of the most abundant oxygenated VOCs, HCHO is one of the major contributors to total VOCs reactivity (Zhang et al., 2012; Steiner et al., 2008). Therefore, it is used to show the model performance on VOCs due to the lack of VOCs observations. Figure 5 shows scatter plots comparing the simulated average daily HCHO and the total VOCs at all 117×117 grids during the study period. It can be seen from the results that HCHO has a high correlation with VOCs, and R2 reaches 0.93. We also compared the simulated HCHO and NO2 with satellite observations (TROPOMI) to further verify the model (Fig. 3). Changes in manuscript: Methodology (Lines 85-89 in the revision): "The satellite-observed NO2 and formaldehyde (HCHO) column number density datasets are from the Sentinel-5 Precursor TROPOspheric Monitoring Instrument (S-5P TROPOMI) (https://scihub.copernicus.eu). Besides, we filter the satellite data under the recommended criteria of QA values greater than 75% for tropospheric NO2 column number density datasets and 50% for HCHO (Apituley, 2018)." Results and discussion (Lines 158-163 in the revision): "To further validate modeled HCHO and NO2, we compared our simulated results with satellite-observed data during pre-lockdown and lockdown periods (Fig. S1). The tropospheric column densities of NO2 and HCHO were calculated by summing their concentrations of 17

vertical layers in the CMAQ model (H. J. Eskes, 2020). The predicted regional distribution of tropospheric column NO2 and HCHO is similar to satellite-observations. Overall, HCHO and NO2 are higher in eastern and northern India than in other regions. And their variation trends from CMAQ and TROPOMI are consistent that NO2 decreases while HCHO increases during the lockdown."

Comments: Are benzene and toluene data available from the monitoring stations which could be included in the analyses? If so these should also be included in view of their health and SOA formation potential. Response: Thanks for the referee's comments. The available observational data for benzene and toluene from the CPCB dataset is extremely limited. For example, Chennai does not have a single monitoring site to provide its hourly observations. In the model simulation, the EDGAR emission inventory does not provide a separate benzene emission and toluene is lumped into ARO1 species in the SAPRC-11 photochemical mechanism (Carter, 2011; Hu et al., 2016). So it is a pity that the observation of benzene and toluene cannot be compared with the model simulation. Besides, our study is not focused on health risks or their specific impact on SOA formation, but on the impact of anthropogenic emission reductions on major air pollutants during the lockdown. Changes in manuscript: No changes were made for this point.

Comments: Choice of scaling factors for emission reductions: The authors make several assumptions and justification for the use of scaling factors for emissions which are valid (see Equations 1 and 2). For example: Ammonia agricultural emissions: Several satellite studies have indicated high ammonia emissions from agriculture and a recent by G.K. Singh, P. Rajeev, D. Paul, et al., Chemical characterization and stable nitrogen isotope composition of nitrogenous component of ambient aerosols, Science of the Total Environment, https://doi.org/10.1016/j.scitotenv.2020.143032 showed that agriculture activities and waste generation are major sources of ammonia. The assumption by the authors that the agricultural emissions do not change between pre-lockdown and during lockdown is not valid for large parts of the India in particular the

Indo-Gangetic Plain because during the pre-lockdown dates farmers were still applying fertilizers to the wheat crops, whereas by last week of March this completely stops. So infact the ammonia and hence ammonium ion source from agriculture is likely stronger in pre-lockdown period and so cannot be treated as constant between both periods. As ammonia is such an important emission for PM2.5 too, this has large implications for the inferences currently drawn by the authors. Response: Thanks for the helpful suggestion from the referee. However, due to the data limitation, we cannot calculate a specific emission reduction ratio for agriculture due to the lockdown on a regional scale. As long as we can get more information, we will further refine the proportion of emission reduction in the lockdown in the follow-up study. Changes in manuscript: No changes were made for this point.

Comments: Ozone production sensitivity indicator: The use of HCHO/NO2 as based on Silman et al 1995 which the authors cite cannot be applied blindly because as noted by the original authors (Silman and He in their JGR paper in 2002) is suitable only for ambient ozone mixing ratios in the range of 80-200 ppb and then again for columns retrieved using satellite data. For ground based data, more robust proxies would be H2O2/ HNO3 or even O3/NOy. Response: Thanks for the referee's comments. As shown in Fig. 7, we change the indicator of O3 sensitivity to NOx and VOCs into O3/NOy and Sillman (1995) suggested the transition value that separate NOx-sensitive and VOC-sensitive locations (O3/NOy= 6-8). According to the value, we can find the most Indian region is NOx-sensitive and the VOC-limited and transition regimes expand during the lockdown because of the reduction of anthropogenic emissions. Changes in manuscript: Results and discussion (Lines 275-280 in the revision): "Figure S5 shows the O3 production sensitivity (O3/NOy) in India during the lockdown, which is considered as an indicator of O3 sensitivity to NOx and VOCs (Sillman, 1995; Sillman and He, 2002). In India, NOx-limited regimes (O3/NOy > 8) are found in vast areas from both Case 1 and Case 2, which was also reported in previous studies (Mahajan et al., 2015). Compared to Case 1, the VOC-limited area (O3/NOy < 6) expands mainly in the northwest and south of India from Case 2 during the lockdown. The transition regimes

(6< O3/NOy < 8) that O3 formation is controlled by both NOx and VOC emissions in the vicinity of the VOC-limited regions."

Comments: In the absence of measured VOC data presented by the authors to validate their model VOC data (note there are no measurements of HCHO presented), the authors should remove this discussion completely or present for each city site the high resolution O3 Vs NOx data from daytime for pre and during lockdown periods. Response: As shown in responses to previous comments, we compared the simulated HCHO with satellite observations (TROPOMI) to further verify the model (Fig. 3). We believe that the discussion is useful to readers, and we acknowledged that more studies are needed to better illustrate the relationship between O3, VOCs, and NOx. Changes in manuscript: Methodology (Lines 85-89 in the revision): "The satellite-observed NO2 and formaldehyde (HCHO) column number density datasets are from the Sentinel-5 Precursor TROPOspheric Monitoring Instrument (S-5P TROPOMI) (https://scihub.copernicus.eu). Besides, we filter the satellite data under the recommended criteria of QA values greater than 75% for tropospheric NO2 column number density datasets and 50% for HCHO (Apituley, 2018)." Results and discussion (Lines 158-163 in the revision): "To further validate modeled HCHO and NO2, we compared our simulated results with satellite-observed data during pre-lockdown and lockdown periods (Fig. S1). The tropospheric column densities of NO2 and HCHO were calculated by summing their concentrations of 17 vertical layers in the CMAQ model (H. J. Eskes, 2020). The predicted regional distribution of tropospheric column NO2 and HCHO is similar to satellite-observations. Overall, HCHO and NO2 are higher in eastern and northern India than in other regions. And their variation trends from CMAQ and TROPOMI are consistent that NO2 decreases while HCHO increases during the lockdown."

Comments: In several instances, the grammar and language also need to be corrected. I recommend the authors to consider the above major concerns to revise and improve the manuscript. Response: As suggested, we made corresponding changes

and improved the grammar and language in the revised manuscript.

Please also note the supplement to this comment:
https://acp.copernicus.org/preprints/acp-2020-903/acp-2020-903-AC3-supplement.pdf

[Figure]

**Fig. 1.**

[Figure]

**Fig. 2.**

[Figure]

**Fig. 3.**

[Figure]

**Fig. 4.**

[Figure]

**Fig. 5.**

[Figure]

**Fig. 6.**

[Figure]

Fig. 7.

**Supplement:**

**Responses to interactive comments**

**Journal: Atmospheric Chemistry and Physics**

**Manuscript ID: acp-2020-903**

**Title: "Impact of reduced anthropogenic emissions during COVID-19 on air quality in India"**

Dear Referee #3,

We appreciate your comments to help improve the manuscript. We tried our best to address your comments and detailed responses and related changes are shown below. Our response is in blue and the modifications in the manuscript are in red.

**Comments:** The paper by Zhang et al. entitled "Impact of reduced anthropogenic emissions during COVID-19 on air quality in India" is on a very relevant and interesting topic which is to use the covid lockdown emission reductions for assessing impacts on air quality over India. Unfortunately, the analyses and interpretation are weak in several places (listed below). There are hardly any new trustworthy insights from this modelling study which have not been reported already by the authors in previous works on the same topic published recently (see Sharma, S., Zhang, M., Anshika, Gao, J., Zhang, H., and Kota, S. H.: Effect of restricted emissions during COVID-19 on air quality in India, Science of The Total Environment, 728, 138878, https://doi.org/10.1016/j.scitotenv.2020.138878, 2020).

**Response:**

We appreciate the comments. However, we don't agree that there are hardly any new trustworthy insights. We comprehensively evaluate the impact of the nationwide lockdown on air quality in India, which also provides reliable recommendations for the improvement of emission reduction policies.

First, we determined the response of air quality in India under the synergetic impacts from the meteorological conditions and anthropogenic emissions during the pre-lockdown and lockdown periods. For instance, we directly quantified the change in air quality during lockdown due to the reduced anthropogenic emissions through differences between Case 1 (without emission reductions) and Case 2 (with emission reductions) during the lockdown. This casts lights on the policy implementation in India, which may help to mitigate air pollution in the future.

Second, we are the first study that explored the impacts of COVID-19 lockdown on Indian air quality on a regional scale. It allows us to figure out the changes of primary and even secondary pollutants during two periods (pre-lockdown and lockdown) and illustrate their differences in urban and rural areas. This could be a great help to formulate the city-level control policy in India.

Third, in atmospheric chemistry, we developed a better understanding of the secondary pollutants formations by investigating their non-linear responses to the precursors' changes during the lockdown. In particular, the sensitivity of $PM_{2.5}$ secondary components (Fig. 6 in the revised manuscript) and the change of spatial distributions of $O_3$ production sensitivity (Fig. S5 in the revised supplement) due to emission changes during the lockdown give us a more in-depth discussion on secondary pollutants.

In the revised manuscript, we added such information to the Introduction to make it clear.

**Changes in manuscript:**

**Introduction (Lines 64-68 in the revision):** "However, the role of meteorological conditions and chemical reactions involving changes in air quality is not clear from these observation-based studies, which only showed the phenomenon of concentration reduction and switch of major primary pollutants mainly in urban cities. Further, the number of monitoring stations in the country is way below the guidelines by the governing bodies and not uniformly distributed, which results in observation data limitations in India (Sahu et al., 2020)."

**Introduction (Lines 69-74 in the revision):** "In this study, the Community Multi-Scale Air Quality (CMAQ) model was used to investigate changes in air pollutants during the pre-lockdown (from February 21, 2020 to March 23, 2020) and lockdown (from March 24, 2020 to April 24, 2020) periods throughout Indian region. We explored the synergetic impacts from the meteorological conditions and anthropogenic emissions during the pre-lockdown and lockdown periods. Besides, we directly quantified the change in air quality during the lockdown due to the reduced anthropogenic emissions by comparing the differences between Case 1 (without emission reductions) and Case 2 (with emission reductions)."

**Conclusion (Lines 309-310 in the revision):** "However, more stringent mitigation measures are needed to achieve effective control of air pollution from secondary air pollutants and their components, particularly in rural areas."

**Comments:** Instead there are even discrepancies from the earlier work based on interpretation of what appears to be the same measured dataset. While in the previous work (Sharma et al., 2020) it was reported that there was a 17% increase in ozone during COVID, in the present work it has been reported that a significant decrease in surface ozone (MDA8 values) occurred, without even clarifying what changed between the two studies except for additional modelling analyses in this study.

**Response:**

We are sorry for being not clear enough. As described in the previous response, these two studies are different in many aspects and we are not aiming to show the same results.

First, the duration of lockdown considered in this study (from March 24 to April 24, 2020) is different from Sharma et al. (from March 15th to April 14, 2020). Second, the variations of MDA8 $O_3$ in this study included all India within the 36-km domain (117×117 grids) (Fig. 1 in the manuscript), while Sharma et al. only focused on urban measurements at 22 cities. Third, this study excluded the influence of meteorology by comparing Case 1 (without emission reductions) and Case 2 (with emission reductions) during the lockdown, while Sharma et al. concluded the increase in $O_3$ by comparing 2020 and the previous three years.

Moreover, the results of our study are consistent with Sharma et al. in the urban areas. In these areas that were under VOC-limited conditions, both studies concluded that $O_3$ increased during the lockdown. In our study, the increase in MDA8 $O_3$ was up to 21%, which was close to Shame et al.

In summary, different research methods and study periods result in different $O_3$ changing ratios, and it is not able to conclude that as discrepancies.

**Changes in manuscript:** Since the differences have been added in the previous comment, no special changes were made for this point.

**Comments:** There are several major issues with the present submission which need to be addressed/clarified for meriting further publication in ACP.

**Response:** We thank the reviewers for the detailed comments below and made necessary changes to the manuscript.

**Comments: Validation of the model used in this work has not been done/described adequately:**

Authors use only 2 m level measurements of temperature and meteorological data and chemical data from 5 monitoring stations operated by the regulatory agency of India located within cities to compare their modelled output.

**Response:**

    We collected all available observations to validate our WRF and CMAQ models. Since the observations are limited in India, it is important to conduct simulation studies like this one to improve our understanding and help design control strategies.

**Changes in manuscript:** No changes were made for this point.

**Comments: Measured chemical data:** The authors present only daily averaged data in the plots (Figures 2 and 3). This would be fine but I could find no details of the original high resolution primary data (presumably available at temporal resolution of few minutes from the analyzers in the monitoring stations) to build confidence in the reader about the trustworthiness of the primary data and its quality assurance. If they could provide such high resolution data for the five stations (even for few days in both periods) for ozone, NO, $NO_2$, $PM_{2.5}$ etc.. with gaps in measurements if any (after all there was a lockdown so maintenance could be difficult), and the calibration data of any of the analyzers, it would go a long way in instilling confidence in the highly averaged data. The reviewer looked up their previous study Sharma et al 2020 which has been cited for detailed description of the primary data and found that this reference did not contain these details and somewhat remarkably the Sharma et al. 2020 paper reported data until April 14th 2020 in that work, was submitted on April 16th, 2020 and accepted on April 19th, 2020. While this does not necessarily suggest that due diligence was not taken as given the nature of topic urgency to publish would have been a factor, the rapid turn-around time and lack of experimental details in the peer reviewed reference cited and which forms the basis of the daily averages does leave room for concern. So the authors should provide the original primary data as a time series for these 5 monitoring stations in the revised supplement along with details of calibration experiments and data quality control followed to allay such potential concerns about the primary measured dataset.

**Response:**

Although higher resolution from some sampling equipment is available, monitoring agencies worldwide only report hourly data. The data from the CPCB database can be downloaded at https://app.cpcbccr.com/ccr/#/caaqm-dashboard-all/caaqm-landing/data and we are not able to post the raw data here due to the CPCB's data access statement. Also, the temporal-resolution of the CMAQ model is hourly. Besides, daily average $PM_{2.5}$ is commonly used in previous studies due to its practical significance and data availability. For $O_3$, the maximum daily 8-hour concentration (MDA8 $O_3$) calculated as running the maximum continuous 8-hour data is used to represent pollution level (Lei et al., 2019). Thus, we used the daily average $PM_{2.5}$ and MDA8 $O_3$ specifically in Figures 2 and 3 in the manuscript.

We are sorry that the validity of the observation data is not clear. The observation data from CPCB has been through quality assurance or quality control programs by establishing strict procedures for sampling, analysis, and calibration before publication (Gurjar et al., 2016). Thus, studies usually use the data directly. In this study, we made additional checks to screen out outliers. For example, a cut-off value of 40 ppb was applied to hourly $O_3$ based on EPA's recommendations (EPA, 2005). For $PM_{2.5}$, abnormally high values of greater than 300 μg $m^{-3}$ were excluded. These explanations were added in different parts of the revised manuscript.

**Changes in manuscript:**

**Methodology (Lines 82-83 in the revision):** "The CPCB database provides data quality assurance (QA) or quality control (QC) programs by establishing strict procedures for sampling, analysis, and calibration (Gurjar et al., 2016)."

**Results and discussion (Lines 150-153 in the revision):** "For $PM_{2.5}$, the averaged mean fractional bias (MFB) (-0.48) and mean fractional error (MFE) (0.61) values met the criteria limits of ±0.6 and 0.75 claimed by the EPA (2007b) in all the five urban cites after excluding some abnormally high values of greater than 300 μg $m^{-3}$. For $O_3$, a cut-off value of 40 ppb is applied, which is based on EPA's recommendations (EPA, 2005)."

**Comments:** Also they should discuss whether data from 5 cities are adequate to make inferences about all of India with same degree of confidence which spans vast rural and countryside regions? It might be advisable to better focus on the 5 cities alone for which they have the data and even there they should acknowledge how data from one monitoring station may be limited for representing air quality of the entire city. In fact, a combination of monitoring station data and satellite data (agreed also can have issues but better than nothing) would be better.

**Response:**

Thanks for the comments. First of all, we agree that 5 cities were maybe inadequate to evaluate the model for the whole of India. However, it is an endless effort and impractical to monitor the whole country. Focusing only on areas with observations is a safe idea, but not a promising one. This actually is the advantage of modelling work and the merit of this study. Besides, when we validate our simulated results in urban areas, we have the confidence to investigate rural and countryside regions. The modelling work is not only to

reproduce what has been observed but more importantly to investigate what is not been observed after enough validation.

If the monitoring station can represent the entire city is another good question for observation experts. In this study, the observation data we compare with the predicted values are an average of multiple sites in each city (Table 1), which can represent the $PM_{2.5}$ and $O_3$ levels of the entire city well. We also added more observations to make the results more representative in the revision (Fig. 1 also added as Fig. 2 in the revised manuscript & Fig. 2 added as Fig. 3 in the revised manuscript). The model performed well at simulating $O_3$, $PM_{2.5}$, CO, and $NO_2$ in these major cities in India (Table 2, also added as Table S7 in the revised supplement).

Thanks to the suggestion of using satellite, we compared model performance with satellite observations (TROPOMI) for HCHO and $NO_2$ (Fig. 3, also revised Fig. S1 in the supplement). The corresponding explanatory statements were added in the Results and discussion section.

**Table 1: Cities and monitoring sites selected for observation data from the CPCB database.**

| City | Monitoring sites |
| --- | --- |
| Hyderabad | Bollaram Industrial Area, Hyderabad - TSPCB |
| | ICRISAT Patancheru, Hyderabad - TSPCB |
| | IDA Pashamylaram, Hyderabad - TSPCB |
| Chennai | Velachery Res. Area, Chennai - CPCB |
| | Alandur Bus Depot, Chennai - CPCB |
| | Manali Village, Chennai - TNPCB |
| Mumbai | Chhatrapati Shivaji Intl. Airport (T2), Mumbai - MPCB |
| | Bandra, Mumbai - MPCB |
| | Borivali East, Mumbai - MPCB |
| | Powai, Mumbai - MPCB |
| | Sion, Mumbai - MPCB |
| Bengaluru | BTM Layout, Bengaluru - CPCB |
| | BWSSB Kadabesanahalli, Bengaluru - CPCB |
| | Silk Board, Bengaluru - KSPCB |
| Delhi | Aya Nagar, Delhi - IMD |
| | Ashok Vihar, Delhi - DPCC |
| | Bawana, Delhi - DPCC |
| | Alipur, Delhi - DPCC |
| | Anand Vihar, Delhi - DPCC |
| | CRRI Mathura Road, Delhi - IMD |
| | DTU, Delhi - CPCB |
| | Dr. Karni Singh Shooting Range, Delhi - DPCC |
| | Dwarka-Sector 8, Delhi - DPCC |
| | ITO, Delhi - CPCB |

Jahangirpuri, Delhi - DPCC

Jawaharlal Nehru Stadium, Delhi - DPCC

Lodhi Road, Delhi - IMD

Major Dhyan Chand National Stadium, Delhi - DPCC

Mandir Marg, Delhi - DPCC

NSIT Dwarka, Delhi - CPCB

Najafgarh, Delhi - DPCC

Nehru Nagar, Delhi - DPCC

North Campus, DU, Delhi - IMD

Okhla Phase-2, Delhi - DPCC

Patparganj, Delhi - DPCC

R K Puram, Delhi - DPCC

Shadipur, Delhi - CPCB

Sri Aurobindo Marg, Delhi - DPCC

Vivek Vihar, Delhi - DPCC

**Table 2: model performance of $O_3$ (ppb), $PM_{2.5}$ (µg m$^{-3}$), CO (ppb), and $NO_2$ (ppb) at Delhi, Mumbai, Chennai, Hyderabad, and Bengaluru (OBS is mean observation; PRE is mean prediction; MFB is mean fractional bias; MFE is mean fractional error; MNB is mean normalized bias; MNE is mean normalized error).**

| Variable | Statistics | Delhi | Mumbai | Chennai | Hyderabad | Bengaluru | ALL | Benchmark |
|---|---|---|---|---|---|---|---|---|
| $O_3$ | OBS | 61.37 | 56.64 | 49.94 | 44.03 | 47.66 | 51.93 | |
| | PRE | 56.86 | 47.06 | 39.32 | 52.56 | 43.58 | 47.88 | |
| | MNB | -0.04 | -0.13 | **-0.20** | **0.20** | -0.07 | -0.05 | ≤±0.15 |
| | MNE | 0.20 | 0.28 | 0.29 | 0.22 | 0.26 | 0.25 | ≤0.30 |
| | MFB | -0.08 | -0.20 | -0.27 | 0.17 | -0.17 | -0.11 | |
| | MFE | 0.21 | 0.32 | 0.36 | 0.18 | 0.33 | 0.28 | |
| $PM_{2.5}$ | OBS | 58.08 | 33.10 | 23.09 | 33.09 | 33.79 | 36.23 | |
| | PRE | 38.10 | 21.46 | 15.57 | 18.28 | 16.29 | 21.94 | |
| | MNB | -0.16 | -0.28 | -0.12 | -0.41 | -0.41 | -0.28 | |
| | MNE | 0.54 | 0.44 | 0.46 | 0.47 | 0.51 | 0.48 | |
| | MFB | -0.40 | -0.46 | -0.30 | -0.59 | **-0.66** | -0.48 | ≤±0.6 |
| | MFE | 0.62 | 0.59 | 0.51 | 0.63 | 0.73 | 0.61 | ≤0.75 |
| $NO_2$ | OBS | 13.87 | 11.17 | 3.74 | 10.60 | 10.68 | 10.01 | |
| | PRE | 7.00 | 9.68 | 4.34 | 3.04 | 8.64 | 6.54 | |
| | MNB | -0.51 | 0.42 | 0.46 | -0.74 | -0.30 | -0.14 | |
| | MNE | 0.74 | 1.34 | 1.11 | 0.86 | 0.89 | 0.99 | |

|   |     |       |       |       |       |       |       |
|---|-----|-------|-------|-------|-------|-------|-------|
|   | MFB | -1.00 | -0.47 | -0.20 | -1.44 | -0.90 | -0.80 |
|   | MFE | 1.13  | 1.18  | 0.82  | 1.51  | 1.19  | 1.17  |
| CO | OBS | 0.69 | 0.65  | 0.45  | 0.38  | 0.74  | 0.58  |
|   | PRE | 0.26  | 0.16  | 0.12  | 0.13  | 0.14  | 0.16  |
|   | MNB | -0.59 | -0.72 | -0.71 | -0.61 | -0.78 | -0.68 |
|   | MNE | 0.59  | 0.72  | 0.71  | 0.61  | 0.78  | 0.68  |
|   | MFB | -0.88 | -1.15 | -1.13 | -0.92 | -1.32 | -1.08 |
|   | MFE | 0.89  | 1.15  | 1.13  | 0.92  | 1.32  | 1.08  |

[Figure]

**Figure 1: Comparison of predicted and observed PM$_{2.5}$ from February 21 to April 24, 2020 in Delhi, Mumbai, Chennai, Hyderabad, and Bengaluru. The unit is µg m$^{-3}$.**

[Figure]

**Figure 2: Comparison of predicted and observed MDA8 O₃ from February 21 to April 24, 2020 in Delhi, Mumbai, Chennai, Hyderabad, and Bengaluru. The unit is ppb.**

[Figure]

**Figure 3: Comparison of the simulated and satellite-observed NO₂ and HCHO column number density before lockdown and during the lockdown in India. The unit is $10^{15}$ molec cm$^{-2}$.**

**Changes in manuscript:**

**Results and discussion (Lines 150-153 in the revision):** "For PM$_{2.5}$, the averaged mean fractional bias (MFB) (-0.48) and mean fractional error (MFE) (0.61) values met the criteria limits of ±0.6 and 0.75 claimed by the EPA (2007b) in all the five urban cites after excluding some abnormally high values of greater than 300 µg m$^{-3}$. For O₃, a cut-off value of 40 ppb is applied, which is based on EPA's recommendations (EPA, 2005)."

**Results and discussion(Lines 154-156 in the revision):** "And averaged MFB (-0.05) and MFE (0.25) values of O₃ also satisfy the benchmarks of ±0.15 and 0.30 set by the EPA (2005) in most of these cities with Chennai and Hyderabad exceeding the limits slightly."

**Results and discussion (Lines 168-172 in the revision):** "Overall, sharp decreases are found in the observed PM$_{2.5}$ in all these cities, and the averaged PM$_{2.5}$ level drops from 43.18 μg m$^{-3}$ to 27.62 μg m$^{-3}$. The mean observed PM$_{2.5}$ concentrations during lockdown are 42.47 μg m$^{-3}$ (Delhi), 24.53 μg m$^{-3}$ (Mumbai), 15.73 μg m$^{-3}$ (Chennai), 31.29 μg m$^{-3}$ (Hyderabad), 24.08 μg m$^{-3}$ (Bengaluru), which are reduced by 41%, 40%, 42%, 10%, and 43% respectively compared with that of the pre-lockdown period. Besides, the observed peak values of PM$_{2.5}$ in each city also decrease appreciably (up to 57%) during the lockdown period."

**Results and discussion (Lines 177-178 in the revision):** "The observed average MDA8 O$_3$ during lockdown is higher than that of pre-lockdown in Delhi (2%), Hyderabad (12%), and Bengaluru (2%)."

**Results and discussion (Lines 182-183 in the revision):** "In contrast, the observed average MDA8 O$_3$ during lockdown is reduced compared with the pre-lockdown period in both Mumbai (-35%) and Chennai (-13%)."

**Methodology (Lines 85-89 in the revision):** "The satellite-observed NO$_2$ and formaldehyde (HCHO) column number density datasets are from the Sentinel-5 Precursor TROPOspheric Monitoring Instrument (S-5P TROPOMI) (https://scihub.copernicus.eu). Besides, we filter the satellite data under the recommended criteria of QA values greater than 75% for tropospheric NO$_2$ column number density datasets and 50% for HCHO (Apituley, 2018)."

**Results and discussion (Lines 158-163 in the revision):** "To further validate modeled HCHO and NO$_2$, we compared our simulated results with satellite-observed data during pre-lockdown and lockdown periods (Fig. S1). The tropospheric column densities of NO$_2$ and HCHO were calculated by summing their concentrations of 17 vertical layers in the CMAQ model (H. J. Eskes, 2020). The predicted regional distribution of tropospheric column NO$_2$ and HCHO is similar to satellite-observations. Overall, HCHO and NO$_2$ are higher in eastern and northern India than in other regions. And their variation trends from CMAQ and TROPOMI are consistent that NO$_2$ decreases while HCHO increases during the lockdown."

**Comments: Validation of model using 2 m level measured meteorological data:** For the kind of modelling investigation the authors are making namely, effect of emission changes on concentrations of pollutants use of only the 2 m level observations without comparison with satellite data, sonde data, mixing layer height data (see ERA5 products) seems to be a major shortcoming. Note that the changes in ventilation coefficient before and during lockdown and the changing season (Spring to Summer) can alone have big impacts on the concentrations.

**Response:**

Thanks for the referee's comments. The weather research and forecasting (WRF) ARW regional model is a high-resolution meteorology model that is widely used in Indian studies (Gevorgyan, 2018; Srinivas et al., 2013; Chawla et al., 2018; Ashrit and Mohandas, 2010). The capability of the WRF model has been validated for providing reliable meteorological inputs to air quality models even under extreme weather events (Zhang et al., 2020; Stella and Agnihotri, 2015; Pattanaik and Rama Rao, 2009; Rajeevan et al., 2010). In the manuscript, we validated the model performance of WRF using observation from the National Climate

Data Center (NCDC). Although only the near-surface meteorological factors are considered, our prediction also shows good performance with the comparison of satellite data (NO$_2$ and HCHO) (Fig. 3, also added as Fig. S1 in the revised supplement). As suggested by the referee, the reanalysis-based ERA5 product produces global atmospheric quantities at 31-km horizontal resolution combining model simulations and observations. However, there are uncertainties such as its high root-mean-square error values in the tropical and subtropical climate zone so that the reliability and applicability of the ERA5 dataset still need to be explored in India with a tropical monsoon climate (Jiang et al., 2020; Kolluru et al., 2020). In follow-up studies, we will consider its application to further validate the WRF model.

As for the impacts of ventilation coefficient and changing season, we added more analysis of the difference in meteorological conditions between pre-lockdown and lockdown periods including temperature (T), relative humidity (RH), planetary boundary layer (PBL) height, the average daily precipitation, and wind fields in Fig. 4 (also added as Fig. S3 in the revised supplement). The explanations about the impacts of these meteorological conditions on Indian air quality are also added in the Results and discussion section.

[Figure]

**Figure 4: Distribution of simulated temperature (T), relative humidity (RH), planetary boundary layer (PBL) height, the average daily precipitation, and wind fields in India before and during the lockdown period. "Case2 - Case1" indicates (Case 2 – Case 1)/Case 1, reported as %.**

**Changes in manuscript:**

**Results and discussion (Lines 199-203 in the revision):** "Variations in near-surface meteorological factors during lockdown also play an important role in $PM_{2.5}$ changes. As is shown in Fig. S3, lower $PM_{2.5}$ in urban areas during lockdown (Fig. 4) may attribute to the decrease of RH and increase of planetary boundary layer (PBL) height, while the decrease of precipitation and WS allows $PM_{2.5}$ to accumulate in some rural areas (Schnell et al., 2018; Le et al., 2020)."

**Results and discussion (Lines 207-211 in the revision):** "Although significant reductions are found in $O_3$ precursor emissions throughout India during the lockdown, the MDA8 $O_3$ has not shown comparable decreasing trends, which is affected by the meteorological conditions such as an increase of temperature and decrease of RH (Fig. S3). Higher temperature speeds up photochemical processes that produce $O_3$, while higher RH reduces them (Chen et al., 2019; Zhao et al., 2017; Ali et al., 2012)."

**Conclusion (Lines 302-303 in the revision):** "It can be concluded that meteorological conditions play an important role in those increases according to the comparison between pre-lockdown (Case 1), and lockdown (Case 2)."

**Comments: Changes in atmospheric chemistry of primary pollutant removal and formation of secondary pollutants:**

Currently the study tends to attribute all the observed concentration changes in pollutants primarily to the emission reductions. However it has been documented elegantly in the following paper: Kroll, J.H., Heald, C.L., Cappa, C.D. et al. The complex chemical effects of COVID-19 shutdowns on air quality. Nat. Chem. 12, 777–779 (2020). https://doi.org/10.1038/s41557-020-0535-z, that several other processes play a big role. The purpose of using a model should be that these effects can be teased out through sensitivity experiments but unfortunately this has not been addressed in current version of the manuscript. For example the authors note that the temperature increased during the lockdown period. A key question is what effect the temperature change and the reduced emission of VOCs (no VOC measurements have been provided at all), NOx and CO would have on the removal rates of primary pollutants and formation of secondary pollutants.

Further have authors identified days when it rained in both pre covid lockdown and during lockdown periods which would cause strong biases for the comparisons.

**Response:**

We showed not only the impact of emission reductions but also that the meteorology conditions and explained that the specific chemical reactions can affect the change of pollutant concentrations. We also added more comprehensive discussions about the effects of chemical processes in the manuscript (see the Results and discussion section).

We utilized the model to explore the comprehensive effects of meteorology including the increased temperature on the change of primary and secondary pollutants by comparing pre-lockdown (Case 1) and lockdown (Case 2) rather than focusing on a single variable. We also investigated the nonlinear relationship between emissions and atmospheric composition. For example, Fig. 8 in the manuscript also showed the relative changes in concentrations of HCHO and $NO_x$ and emissions of HCHO and $NO_x$ in eight major cities

of India in Case 2 to Case 1 and proved that the concentration of $NO_x$ is appreciably highly sensitive to a primary $NO_x$ emission reduction compared with HCHO.

In specific, though we can't obtain a reliable VOC observation dataset, the high correlation between HCHO and the total VOCs is shown in Fig. 5 below, and HCHO has been further validated by satellite-observations (Fig. 3, also added as Fig. S1 in the revised supplement).

We also added discussion on the effects of variations in precipitation during the lockdown on $PM_{2.5}$ and MDA8 $O_3$ changes in the Results and discussion results. In addition to the regional change of precipitation shown in Fig. 4 (also added as Fig. S3 in the revised supplement), we also added the daily average precipitation figures in these 5 major cities from February 21 to April 24, 2020 (Fig. 6 in the response). On average, the precipitation in India is relatively low from the pre-lockdown to lockdown periods (lower than 1 mm for each city). Although the few rainy days (such as March 5, 2020) may promote $PM_{2.5}$ removal, generally, it has little impact on the comparison of overall air quality before and during the lockdown.

[Figure]

**Figure 5: Scatter plots comparing the simulated average daily HCHO and the total VOCs at all 117×117 grids from February 21 to April 24, 2020.**

[Figure]

**Figure 6: The predicted average daily precipitation from February 21 to April 24, 2020 in Delhi, Mumbai, Chennai, Hyderabad, and Bengaluru. The unit is mm.**

**Changes in manuscript:**

**Results and discussion (Lines 199-203 in the revision):** "Variations in near-surface meteorological factors during lockdown also play an important role in $PM_{2.5}$ changes. As is shown in Fig. S3, lower $PM_{2.5}$ in urban areas during lockdown (Fig. 4) may attribute to the decrease of RH and increase of planetary boundary layer (PBL) height, while the decrease of precipitation and WS allows $PM_{2.5}$ to accumulate in some rural areas (Schnell et al., 2018; Le et al., 2020)."

**Results and discussion (Lines 241-242 in the revision):** "Besides, the reduction of $NO_x$ may lead to an increase of SOA offsetting some of the influence caused by the reduction in VOC emissions (Kroll et al., 2020)."

**Comments: Existing inadequacies in VOC emission inventories and modelled ozone simulations over India:** While the authors are using pre lockdown and lockdown periods for comparison, it is a fact that of all emission inventories, VOC emissions are the most poorly constrained due to the absence of in-situ VOC data over many regions in India. A generic problem also seen is the tendency for overestimation of ozone by models over the Indian region. This suggests that the basic reactant mixture and chemistry are still inadequate for modelling ozone and secondary pollutant formation accurately over India. So how can one be sure that the changed chemical mixture between pre-lockdown and during lockdown are not skewed by these gaps in our basic underntanding? While it would be unfair to hold the authors to solve all these issues, one does expect that the limitations and existing issues are duly acknowledged in the work instead of making highly speculative and prescriptive measures for air quality mitigation based on such modelling results.

**Response:**

Thanks for pointing out this. We also acknowledge that there are still many limitations and deficiencies in the VOC emission inventories, and the simulation of $O_3$ is usually over-estimated (Kota et al., 2018; Hu et al., 2015). Much more needs to be done regarding emissions and chemical reactions to better simulate $O_3$ in India. But there is no denying that the simulation results of this model are acceptable compared with the standards recommended by EPA (EPA, 2005, 2007a). In the future, we will continuously apply new information to improve our modelling results once they are available.

**Changes in manuscript:**

**Results and discussion (Lines 163-164 in the revision):** "We also acknowledge that the uncertainty of emission inventory and chemical mechanism in the modelling may affect the simulated results (Dominutti et al., 2020; Kitayama et al., 2019)."

**Comments:** Use of formaldehye for constraining VOC emissions where a large number of more reactive primary VOC emissions occur should also be discussed and clarified. Trusting the formaldehye from the model in absence of in-situ formaldehye measurements to compare with or even satellite or columnar measurements which have been reported from India is recommended.

**Response:**

As one of the most abundant oxygenated VOCs, HCHO is one of the major contributors to total VOCs reactivity (Zhang et al., 2012; Steiner et al., 2008). Therefore, it is used to show the model performance on VOCs due to the lack of VOCs observations. Figure 5 (added as Fig.S4 in the revised supplement) shows scatter plots comparing the simulated average daily HCHO and the total VOCs at all 117×117 grids during the study period. It can be seen from the results that HCHO has a high correlation with VOCs, and $R^2$ reaches 0.93. We also compared the simulated HCHO and $NO_2$ with satellite observations (TROPOMI) to further verify the model (Fig. 3, also Fig. S1 in the revised supplement).

**Changes in manuscript:**

**Methodology (Lines 85-89 in the revision):** "The satellite-observed $NO_2$ and formaldehyde (HCHO) column number density datasets are from the Sentinel-5 Precursor TROPOspheric Monitoring Instrument (S-5P TROPOMI) (https://scihub.copernicus.eu). Besides, we filter the satellite data under the recommended criteria of QA values greater than 75% for tropospheric $NO_2$ column number density datasets and 50% for HCHO (Apituley, 2018)."

**Results and discussion (Lines 158-163 in the revision):** "To further validate modeled HCHO and $NO_2$, we compared our simulated results with satellite-observed data during pre-lockdown and lockdown periods (Fig. S1). The tropospheric column densities of $NO_2$ and HCHO were calculated by summing their concentrations of 17 vertical layers in the CMAQ model (H. J. Eskes, 2020). The predicted regional distribution of tropospheric column $NO_2$ and HCHO is similar to satellite-observations. Overall, HCHO and $NO_2$ are higher in eastern and northern India than in other regions. And their variation trends from CMAQ and TROPOMI are consistent that $NO_2$ decreases while HCHO increases during the lockdown."

**Comments:** Are benzene and toluene data available from the monitoring stations which could be included in the analyses? If so these should also be included in view of their health and SOA formation potential.

**Response:**

Thanks for the referee's comments. The available observational data for benzene and toluene from the CPCB dataset is extremely limited. For example, Chennai does not have a single monitoring site to provide its hourly observations. In the model simulation, the EDGAR emission inventory does not provide a separate benzene emission and toluene is lumped into ARO1 species in the SAPRC-11 photochemical mechanism (Carter, 2011; Hu et al., 2016). So it is a pity that the observation of benzene and toluene cannot be compared with the model simulation. Besides, our study is not focused on health risks or their specific impact on SOA formation, but on the impact of anthropogenic emission reductions on major air pollutants during the lockdown.

**Changes in manuscript:** No changes were made for this point.

**Comments: Choice of scaling factors for emission reductions:** The authors make several assumptions and justification for the use of scaling factors for emissions which are valid (see Equations 1 and 2).

For example:

Ammonia agricultural emissions: Several satellite studies have indicated high ammonia emissions from agriculture and a recent by G.K. Singh, P. Rajeev, D. Paul, et al., Chemical characterization and stable nitrogen isotope composition of nitrogenous component of ambient aerosols, Science of the Total Environment, https://doi.org/10.1016/j.scitotenv.2020.143032 showed that agriculture activities and waste generation are major sources of ammonia. The assumption by the authors that the agricultural emissions do not change between pre-lockdown and during lockdown is not valid for large parts of the India in particular the Indo-Gangetic Plain because during the pre-lockdown dates farmers were still applying fertilizers to the wheat crops, whereas by last week of March this completely stops. So infact the ammonia and hence ammonium ion source from agriculture is likely stronger in pre-lockdown period and so cannot be treated as constant between both periods. As ammonia is such an important emission for $PM_{2.5}$ too, this has large implications for the inferences currently drawn by the authors.

**Response:**

Thanks for the helpful suggestion from the referee. However, due to the data limitation, we cannot calculate a specific emission reduction ratio for agriculture due to the lockdown on a regional scale. As long as we can get more information, we will further refine the proportion of emission reduction in the lockdown in the follow-up study.

**Changes in manuscript:** No changes were made for this point.

**Comments:** Ozone production sensitivity indicator: The use of $HCHO/NO_2$ as based on Silman et al 1995 which the authors cite cannot be applied blindly because as noted by the original authors (Silman and He in their JGR paper in 2002) is suitable only for ambient ozone mixing ratios in the range of 80-200 ppb and then

again for columns retrieved using satellite data. For ground based data, more robust proxies would be $H_2O_2$/ $HNO_3$ or even $O_3/NO_y$.

**Response:**

Thanks for the referee's comments. As shown in Fig. 7 (revised Fig. S5 in the supplement), we change the indicator of $O_3$ sensitivity to $NO_x$ and VOCs into $O_3/NO_y$ and Sillman (1995) suggested the transition value that separate $NO_x$-sensitive and VOC-sensitive locations ($O_3/NO_y$= 6-8). According to the value, we can find the most Indian region is $NO_x$-sensitive and the VOC-limited and transition regimes expand during the lockdown because of the reduction of anthropogenic emissions.

[Figure]

**Figure 7: Spatial distributions of O₃ production sensitivity in India from March 24 to April 24, 2020.**

**Changes in manuscript:**

**Results and discussion (Lines 275-280 in the revision):** "Figure S5 shows the $O_3$ production sensitivity ($O_3/NO_y$) in India during the lockdown, which is considered as an indicator of $O_3$ sensitivity to $NO_x$ and VOCs (Sillman, 1995; Sillman and He, 2002). In India, $NO_x$-limited regimes ($O_3/NO_y > 8$) are found in vast areas from both Case 1 and Case 2, which was also reported in previous studies (Mahajan et al., 2015). Compared to Case 1, the VOC-limited area ($O_3/NO_y < 6$) expands mainly in the northwest and south of India from Case 2 during the lockdown. The transition regimes ($6< O_3/NO_y < 8$) that $O_3$ formation is controlled by both $NO_x$ and VOC emissions in the vicinity of the VOC-limited regions."

**Comments:** In the absence of measured VOC data presented by the authors to validate their model VOC data (note there are no measurements of HCHO presented), the authors should remove this discussion completely or present for each city site the high resolution $O_3$ Vs $NO_x$ data from daytime for pre and during lockdown periods.

**Response:**

As shown in responses to previous comments, we compared the simulated HCHO with satellite observations (TROPOMI) to further verify the model (Fig. 3, also revised Fig. S1 in the supplement). We believe that the discussion is useful to readers, and we acknowledged that more studies are needed to better illustrate the relationship between $O_3$, VOCs, and $NO_x$.

**Changes in manuscript:**

**Methodology (Lines 85-89 in the revision):** "The satellite-observed $NO_2$ and formaldehyde (HCHO) column number density datasets are from the Sentinel-5 Precursor TROPOspheric Monitoring Instrument (S-5P TROPOMI) (https://scihub.copernicus.eu). Besides, we filter the satellite data under the recommended criteria of QA values greater than 75% for tropospheric $NO_2$ column number density datasets and 50% for HCHO (Apituley, 2018)."

**Results and discussion (Lines 158-163 in the revision):** "To further validate modeled HCHO and $NO_2$, we compared our simulated results with satellite-observed data during pre-lockdown and lockdown periods (Fig. S1). The tropospheric column densities of $NO_2$ and HCHO were calculated by summing their concentrations of 17 vertical layers in the CMAQ model (H. J. Eskes, 2020). The predicted regional distribution of tropospheric column $NO_2$ and HCHO is similar to satellite-observations. Overall, HCHO and $NO_2$ are higher in eastern and northern India than in other regions. And their variation trends from CMAQ and TROPOMI are consistent that $NO_2$ decreases while HCHO increases during the lockdown."

**Comments:** In several instances, the grammar and language also need to be corrected. I recommend the authors to consider the above major concerns to revise and improve the manuscript.

**Response:** As suggested, we made corresponding changes and improved the grammar and language in the revised manuscript.

**Reference**

Ali, K., Inamdar, S. R., Beig, G., Ghude, S., and Peshin, S.: Surface ozone scenario at Pune and Delhi during the decade of 1990s, Journal of Earth System Science, 121, 373-383, https://doi.org/10.1007/s12040-012-0170-1, 2012.

Apituley, A., Pedergnana, M., Sneep, M., Pepijn Veefkind, J., Loyola, D., Landgraf, J., Borsdorff, T.: Sentinel-5 Precursor/TROPOMI Level 2 Product User Manual Carbon Monoxide, . Royal Netherlands Meteorological Institute., 2018.

Ashrit, R., and Mohandas, S.: Mesoscale model forecast verification during monsoon 2008, Journal of Earth System Science, 119, 417-446, https://doi.org/10.1007/s12040-010-0030-9, 2010.

Carter, W. P. L.: SAPRC Atmospheric Chemical Mechanisms and VOC Reactivity Scales, http://www.cert.ucr.edu/⁓carter/SAPRC/, 2011.

Chawla, I., Osuri, K. K., Mujumdar, P. P., and Niyogi, D.: Assessment of the Weather Research and Forecasting (WRF) model for simulation of extreme rainfall events in the upper Ganga Basin, Hydrol. Earth Syst. Sci., 22, https://doi.org/1095-1117, 10.5194/hess-22-1095-2018, 2018.

Chen, Z., Zhuang, Y., Xie, X., Chen, D., Cheng, N., Yang, L., and Li, R.: Understanding long-term variations of meteorological influences on ground ozone concentrations in Beijing During 2006–2016, Environmental Pollution, 245, 29-37, https://doi.org/10.1016/j.envpol.2018.10.117, 2019.

Dominutti, P., Nogueira, T., Fornaro, A., and Borbon, A.: One decade of VOCs measurements in São Paulo

megacity: Composition, variability, and emission evaluation in a biofuel usage context, Science of The Total Environment, 738, 139790, https://doi.org/10.1016/j.scitotenv.2020.139790, 2020.

EPA: Guidance on the Use of Models and Other Analyses in Attainment Demonstrations for the 8-hour Ozone NAAQS, 2005.

EPA, U. E. P. A., Office of Air Quality Planning Standards: Guidance on the use of models and other analyses for demonstrating attainment of air quality goals for ozone, $PM_{2.5}$, and regional haze, 2007b.

Gevorgyan, A.: Convection-Permitting Simulation of a Heavy Rainfall Event in Armenia Using the WRF Model, Journal of Geophysical Research: Atmospheres, 123, 11,008-011,029, https://doi.org/10.1029/2017JD028247, 2018.

Gurjar, B. R., Ravindra, K., and Nagpure, A. S.: Air pollution trends over Indian megacities and their local-to-global implications, Atmospheric Environment, 142, 475-495, https://doi.org/10.1016/j.atmosenv.2016.06.030, 2016.

S5P Mission Performance Centre Nitrogen Dioxide [L2__NO2___] Readme: https://sentinel.esa.int/documents/247904/3541451/Sentinel-5P-Nitrogen-Dioxide-Level-2-Product-Readme-File, 2020.

Hu, J., Wu, L., Zheng, B., Zhang, Q., He, K., Chang, Q., Li, X., Yang, F., Ying, Q., and Zhang, H.: Source contributions and regional transport of primary particulate matter in China, Environmental Pollution, 207, 31-42, https://doi.org/10.1016/j.envpol.2015.08.037, 2015.

Hu, J., Chen, J., Ying, Q., and Zhang, H.: One-year simulation of ozone and particulate matter in China using WRF/CMAQ modeling system, Atmospheric Chemistry and Physics, 16, 10333-10350, https://doi.org/10.5194/acp-16-10333-2016, 2016.

Jiang, Q., Li, W., Fan, Z., He, X., Sun, W., Chen, S., Wen, J., Gao, J., and Wang, J.: Evaluation of the ERA5 reanalysis precipitation dataset over Chinese Mainland, Journal of Hydrology, 125660, https://doi.org/10.1016/j.jhydrol.2020.125660, 2020.

Kitayama, K., Morino, Y., Yamaji, K., and Chatani, S.: Uncertainties in $O_3$ concentrations simulated by CMAQ over Japan using four chemical mechanisms, Atmospheric Environment, 198, 448-462, https://doi.org/10.1016/j.atmosenv.2018.11.003, 2019.

Kolluru, V., Kolluru, S., and Konkathi, P.: Evaluation and integration of reanalysis rainfall products under contrasting climatic conditions in India, Atmospheric Research, 246, 105121, https://doi.org/10.1016/j.atmosres.2020.105121, 2020.

Kota, S. H., Guo, H., Myllyvirta, L., Hu, J., Sahu, S. K., Garaga, R., Ying, Q., Gao, A., Dahiya, S., Wang, Y., and Zhang, H.: Year-long simulation of gaseous and particulate air pollutants in India, Atmospheric Environment, 180, 244-255, https://doi.org/10.1016/j.atmosenv.2018.03.003, 2018.

Kroll, J. H., Heald, C. L., Cappa, C. D., Farmer, D. K., Fry, J. L., Murphy, J. G., and Steiner, A. L.: The complex chemical effects of COVID-19 shutdowns on air quality, Nature Chemistry, 12, 777-779, https://doi.org/10.1038/s41557-020-0535-z, 2020.

Le, T., Wang, Y., Liu, L., Yang, J., Yung, Y. L., Li, G., and Seinfeld, J. H.: Unexpected air pollution with

marked emission reductions during the COVID-19 outbreak in China, Science, 369, 702, https://doi.org/10.1126/science.abb7431, 2020.

Lei, R., Talbot, R., Wang, Y., Wang, S.-C., and Estes, M.: Surface MDA8 ozone variability during cold front events over the contiguous United States during 2003–2017, Atmospheric Environment, 213, 359-366, https://doi.org/10.1016/j.atmosenv.2019.06.003, 2019.

Mahajan, A. S., De Smedt, I., Biswas, M. S., Ghude, S., Fadnavis, S., Roy, C., and van Roozendael, M.: Inter-annual variations in satellite observations of nitrogen dioxide and formaldehyde over India, Atmospheric Environment, 116, 194-201, https://doi.org/10.1016/j.atmosenv.2015.06.004, 2015.

Pattanaik, D. R., and Rama Rao, Y. V.: Track prediction of very severe cyclone 'Nargis' using high resolution weather research forecasting (WRF) model, Journal of Earth System Science, 118, 309, https://doi.org/10.1007/s12040-009-0031-8, 2009.

Rajeevan, M., Kesarkar, A., Thampi, S. B., Rao, T. N., Radhakrishna, B., and Rajasekhar, M.: Sensitivity of WRF cloud microphysics to simulations of a severe thunderstorm event over Southeast India, Ann. Geophys., 28, 603-619, https://doi.org/10.5194/angeo-28-603-2010, 2010.

Sahu, S. K., Sharma, S., Zhang, H., Chejarla, V., Guo, H., Hu, J., Ying, Q., Xing, J., and Kota, S. H.: Estimating ground level $PM_{2.5}$ concentrations and associated health risk in India using satellite based AOD and WRF predicted meteorological parameters, Chemosphere, 255, https://doi.org/10.1016/j.chemosphere.2020.126969, 2020.

Schnell, J. L., Naik, V., Horowitz, L. W., Paulot, F., Mao, J., Ginoux, P., Zhao, M., and Ram, K.: Exploring the relationship between surface $PM_{2.5}$ and meteorology in Northern India, Atmospheric Chemistry and Physics, 18, 10157-10175, https://doi.org/10.5194/acp-18-10157-2018, 2018.

Sillman, S.: The use of $NO_y$, $H_2O_2$, and $HNO_3$ as indicators for ozone-$NO_x$-hydrocarbon sensitivity in urban locations, Journal of Geophysical Research: Atmospheres, 100, 14175-14188, https://doi.org/10.1029/94JD02953, 1995.

Sillman, S., and He, D.: Some theoretical results concerning $O_3$-$NO_x$-VOC chemistry and NOx-VOC indicators, Journal of Geophysical Research: Atmospheres, 107, ACH 26-21-ACH 26-15, https://doi.org/10.1029/2001JD001123, 2002.

Srinivas, C. V., Hariprasad, D., Bhaskar Rao, D. V., Anjaneyulu, Y., Baskaran, R., and Venkatraman, B.: Simulation of the Indian summer monsoon regional climate using advanced research WRF model, International Journal of Climatology, 33, 1195-1210, https://doi.org/10.1002/joc.3505, 2013.

Steiner, A. L., Cohen, R. C., Harley, R. A., Tonse, S., Millet, D. B., Schade, G. W., and Goldstein, A. H.: VOC reactivity in central California: comparing an air quality model to ground-based measurements, Atmospheric Chemistry and Physics, 8, 351-368, https://doi.org/10.5194/acp-8-351-2008, 2008.

Stella, S., and Agnihotri, G.: Simulation of Severe Convective Weather Events over Southern India Using WRF Model, in: High-Impact Weather Events over the SAARC Region, edited by: Ray, K., Mohapatra, M., Bandyopadhyay, B. K., and Rathore, L. S., Springer International Publishing, Cham, 73-83, 2015.

Zhang, Q., Shao, M., Li, Y., Lu, S. H., Yuan, B., and Chen, W. T.: Increase of ambient formaldehyde in Beijing

and its implication for VOC reactivity, Chinese Chemical Letters, 23, 1059-1062, https://doi.org/10.1016/j.cclet.2012.06.015, 2012.

Zhang, T., Zhao, C., Gong, C., and Pu, Z.: Simulation of Wind Speed Based on Different Driving Datasets and Parameterization Schemes Near Dunhuang Wind Farms in Northwest of China, Atmosphere, 11, 647, 2020.

Zhao, B., Wu, W., Wang, S., Xing, J., Chang, X., Liou, K.-N., Jiang, J. H., Gu, Y., Jang, C., Fu, J. S., Zhu, Y., Wang, J., Lin, Y., and Hao, J.: A modeling study of the nonlinear response of fine particles to air pollutant emissions in the Beijing–Tianjin–Hebei region, Atmospheric Chemistry and Physics, 17, 12031-12050, https://doi.org/10.5194/acp-17-12031-2017, 2017.

---

## Author Response (AR2)

**Point-by-point responses**

**Journal: Atmospheric Chemistry and Physics**

**Manuscript ID: acp-2020-903**

**Title: "Impact of reduced anthropogenic emissions during COVID-19 on air quality in India"**

Dear Editor,

Thank you for deciding to accept our paper. We have substantially revised our manuscript after reading all the comments. Our responses are in blue and the modifications in the manuscript are in red.

**Comments:** L20: clarify what you mean with observed: calculated or measured or both? Clarify what you mean with decreasing rates: I think concentration reductions, correct?

**Response**: We are sorry for not being clear enough. The significant concentration reductions of $PM_{2.5}$ and its major components are CMAQ model predicted results. Accordingly, we changed "observed" into "predicted" in the revised manuscript. And "decreasing rates" means concentration reductions exactly. We also clarified the meaning in the revised manuscript.

**Changes in manuscript:**

**Abstract (Lines 19-21 in the revision):** "Significant reductions of $PM_{2.5}$ concentration and its major components are predicted, especially for secondary inorganic aerosols that are up to 92%, 57%, and 79% for nitrate ($NO_3^-$), sulfate ($SO_4^{2-}$), ammonium ($NH_4^+$), respectively."

**Comments:** L23: Is this a place give some more detail on which emission reductions where more important (NOx or VOC, and link to the VOC: NOx regimes.

**Response**: Thanks for the suggestion. We added more explanations about the relative change of $NO_x$ and VOC concentrations and the $O_3$ sensitive regimes.

**Changes in manuscript:**

**Abstract (Lines 21-23 in the revision):** "On average, the MDA8 $O_3$ also decreases 15% during the lockdown period although it increases sparsely in some VOC-limited urban locations, which is mainly due to the more significant reduction of $NO_x$ than VOCs."

**Comments:** L88: Please explain in one sentence what these 75 and 50 % QA effectively do, and how many or % datasets are discarded due to this.

**Response**: We are sorry for not being clear enough. As suggested by TROPOMI (Apituley, 2018), the quality assurance (QA) values (0.75 for $NO_2$ and 0.5 for HCHO) were used to filter the source data to exclude the interferences such as clouds and snow/ice. As a result, a total of 0.4% and 2.4% $NO_2$ and HCHO data were removed from our study. We made corresponding changes in the revised manuscript.

**Changes in manuscript:**

**Methodology (Lines 87-89 in the revision):** "Besides, we effectively removed the pixels with a QA value less than 0.75 for $NO_2$ tropospheric column density and 0.5 for HCHO from the datasets to exclude the interferences such as clouds and snow/ice (Apituley, 2018)."

**Comments:** L121-125: Try to find a more simple way to say which reductions were applied to very polluting, medium polluting, and low polluting industries.

**Response**: Thanks for the comments. We improved the presentation of industrial emissions reduction calculations. To make it more transparent, we also changed "red, orange, and green industries" into "very polluting (VP), medium polluting (MP), and low polluting (LP) industries." The two tables (Table S4 and Table 1) involved have also been revised accordingly. We made corresponding changes in the revised manuscript.

**Changes in manuscript:**

**Methodology (Lines 114-126 in the revision):** "For the industrial sector, we classify the Indian industries into three different classes based on the degree of air pollution caused (https://www.indianmirror.com/indian-industries/environment.html) (Table S4) including very polluting (VP), medium polluting (MP), and low polluting (LP) industries. The Pollution Index (PI) of any industry is a number ranging from 0 to 100, and the increasing value of PI denotes the rising degree of pollution load from the industry. Besides, CPCB, State Pollution Control Boards (SPCBs), and the Ministry of Environment, Forest and Climate Change (MoEFCC) have finalized the criteria on the range of PI for the purpose of categorization of the industrial sector (https://pib.gov.in/newsite/printrelease.aspx?relid=137373) (Table 1).

Based on the above definition of the VP, MP, and LP industry, the emissions before lockdown can be expressed as:

$$E_1=N_{VP-pre}\times S_{VP}+N_{MP-pre}\times S_{MP}+N_{LP-pre}\times S_{LP} \ , \qquad\qquad (1)$$

where $S_{VP}$, $S_{MP}$, and $S_{LP}$ are 1, 0.6, and 0.4 as the assigned scores, and $N_{VP-pre}$, $N_{MP-pre}$, and $N_{LP-pre}$ are the number of each category industry during pre-lockdown. Similarly, the emissions during the lockdown are as follows:

$$E_2=N_{VP-lock}\times S_{VP}+N_{MP-lock}\times S_{MP}+N_{LP-lock}\times S_{LP} \ , \qquad\qquad (2)$$

where $N_{VP-lock}$, $N_{MP-lock}$, and $N_{LP-lock}$ are the number of functioning industries during the lockdown."

**Comments:** L141: Either define here what you mean with acceptable or rather not mention this. Later you mention 2 % for GE/EWS.

**Response**: Thanks for pointing out this. We deleted the inaccurate statements in the revised manuscript.

**Changes in manuscript:**

**Results and discussion (Lines 141 in the revision):** "In general, the WRF model performance is similar to previous studies in India (Kota et al., 2018)."

**Comments:** L151: How does EPA (2007) pertain to Indian cities. Needs further explanation.

**Response**: Thanks for the comments. The $PM_{2.5}$ Criteria from EPA (2007) are commonly used for validating air quality model performance in India, such as Mohan and Gupta (2018), Kota et al. (2018), and so on. We make further explanations in the revised manuscript.

**Changes in manuscript:**

**Results and discussion (Lines 150-153 in the revision):** "For $PM_{2.5}$, after excluding some abnormally high values of greater than 300 µg m$^{-3}$, the averaged mean fractional bias (MFB) (-0.48) and mean fractional error (MFE) (0.61) values in all the five urban cites met the criteria limits of ±0.6 and 0.75 claimed by the EPA (2007). And the recommended criteria are commonly used for validating air quality model performance in the Indian region (Mohan and Gupta, 2018; Kota et al., 2018)."

**Comments:** L160: summing up of concentrations doesn't give columns- please explain better (equation?).

**Response**: Thanks for the comments. We added Eq. (4) in the revised manuscript to clarify the calculation of the tropospheric column densities of $NO_2$ and HCHO.

**Changes in manuscript:**

**Results and discussion (Lines 160-164 in the revision):** "The CMAQ predicted vertical column densities (VCD) of tropospheric $NO_2$ and HCHO were calculated using Eq. (4) (H. J. Eskes, 2020).

$$VCD = \sum_{i=1}^{n} C_i \times H_i \times \alpha , \tag{4}$$

where n equals 17 as the number of vertical layers in the model (with the highest layer height of ~10 km), $C_i$ means species concentration (ppm), $H_i$ represents each layer height (m), and $\alpha$ is the coefficient for converting units from ppm to molec $cm^{-2}$."

**Comments:** L176: here and at various other spots. Careful use of the word trend. In this case you probably mean a tendency. In other cases you discuss a step-change rather than a trend. Check.

**Response**: Thanks for pointing out this. In this sentence, we meant a tendency. But elsewhere, there was a misuse of the word trend. We made corresponding changes in the revised manuscript.

**Changes in manuscript:**

**Results and discussion (Lines 191-193 in the revision):** "Generally, decreases of key pollutants including particulate matter with an aerodynamic diameter of less than 10 μm ($PM_{10}$) (-16%), $PM_{2.5}$ (-26%), MDA8 $O_3$ (-11%), $NO_2$ (-50%), and sulfur dioxide ($SO_2$) (-14%) are calculated across India."

**Results and discussion (Lines 198-199 in the revision):** "However, increases in these key pollutants are found mainly in the northeastern, eastern, and parts of southern India."

**Results and discussion (Lines 204 in the revision):** "However, increases of $PM_{2.5}$ (~20%) are observed in the far-flung northeastern part of India."

**Results and discussion (Lines 205-208 in the revision):** "As is shown in Fig. S3, lower $PM_{2.5}$ in urban areas during lockdown (Fig. 4) may be attributed to the decrease of RH and increase of planetary boundary layer (PBL) height, while the decrease of precipitation and WS allows $PM_{2.5}$ to accumulate in some rural areas (Schnell et al., 2018; Le et al., 2020)."

**Results and discussion (Lines 212-214 in the revision):** "Although significant reductions are found in $O_3$ precursor emissions throughout India during the lockdown, the MDA8 $O_3$ has not shown a comparable decrease, which is affected by meteorological conditions such as an increase of temperature and decrease of RH (Fig. S3)."

**Conclusion (Lines 307-308 in the revision):** "Compared with pre-lockdown, observed $PM_{2.5}$ during the lockdown in Delhi, Mumbai, Chennai, Hyderabad, and Bengaluru shows an overall decrease."

**Comments:** L185: Interesting to know, but explain how that affects the lockdown which was supposedly everywhere. Was it more stringently implied in Mumbai than elsewhere, it may have an implication for your model assumptions.

**Response**:

We are sorry for not being clear enough. Though nationwide lockdown was imposed, more stringent lockdown measures were implemented in major cities and the worst-hit areas in India (https://www.hindustantimes.com/india-news/lockdown-5-0-these-13-cities-will-see-stricter-rules-more-monitoring/story-FNB1TTTIwBqgILlvCbhQUO.html). Besides, more strict lockdown measures were supposed to be implemented in Mumbai, which accounted for more than a fifth of infections in India (Mukherjee, 2020).

For our model assumption, we reduced the anthropogenic emissions by emission sources during the lockdown, not by regions. Mumbai was significantly affected due to its high industrialization with large emissions reduction in industrial and transportation emission sources. We made more corresponding explanations in the revised manuscript.

**Changes in manuscript:**

**Results and discussion (Lines 187-190 in the revision):** "This could be caused by a much larger reduction in emissions as Mumbai and Chennai with high urbanization and industrialization are the most affected areas. In specific, more stringent lockdown measures may be implemented in Mumbai than we assumed, which accounted for more than a fifth of infections in India (Mukherjee, 2020). "

**Comments:** L187: carefully check whether you really mean a decreasing trend- i.e. a trend that is getting less...Is it a trend at all?

**Response**: Thanks for pointing out this. The results only showed a decrease, not a decreasing trend. We made corresponding changes in the revised manuscript.

**Changes in manuscript:**

**Results and discussion (Lines 191-193 in the revision):** "Generally, decreases of key pollutants including particulate matter with an aerodynamic diameter of less than 10 μm ($PM_{10}$) (-16%), $PM_{2.5}$ (-26%), MDA8 $O_3$ (-11%), $NO_2$ (-50%), and sulfur dioxide ($SO_2$) (-14%) are calculated across India."

**Comments:** L201: attribute=>be attributed.

**Response**: Thanks for the comments. The corresponding changes have been made in the revised manuscript.

**Changes in manuscript:**

**Results and discussion (Lines 205-208 in the revision):** "As is shown in Fig. S3, lower $PM_{2.5}$ in urban areas during lockdown (Fig. 4) may be attributed to the decrease of RH and increase of planetary boundary layer (PBL) height, while the decrease of precipitation and WS allows $PM_{2.5}$ to accumulate in some rural areas (Schnell et al., 2018; Le et al., 2020)."

**Comments:** L209: decreasing trends? Just decreases?

**Response**: Thanks for the comments. We meant just decreases here. The corresponding changes have been made in the revised manuscript.

**Changes in manuscript:**

**Results and discussion (Lines 212-214 in the revision):** "Although significant reductions are found in $O_3$ precursor emissions throughout India during the lockdown, the MDA8 $O_3$ has not shown a comparable decrease, which is affected by meteorological conditions such as an increase of temperature and decrease of RH (Fig. S3)."

**Comments:** L216: duration of the lockdown+> effective implementation of the lockdown?

**Response**: We are sorry for not being clear enough. In our study, the comparison of the lockdown duration with previous studies aimed to explain the difference between our results and those of previous studies such as Chauhan and Singh (2020), Mahato et al. (2020), and Kumari and Toshniwal (2020). Besides, the lockdown duration can indicate the effectiveness of the lockdown to some extent because there was a relaxation period in the later lockdown (after April 15, 2020), when traffic flow increased (Kumar, 2020). The relaxation period was included in our study, which led to less $PM_{2.5}$ reduction than the previous studies that mainly focused on the first phase of lockdown (from March 24, 2020 to April 15, 2020).

**Changes in manuscript:**

**Results and discussion (Lines 221-223 in the revision):** "These differences may be caused by the considered duration of lockdown period. The later lockdown period (after April 15, 2020) is concerned

in our study when there is an increase in traffic flow and some relaxation of lockdown measures (Kumar, 2020)."

**Comments:** L225: improve English

**Response**: Thanks for the comments. We improved the corresponding statement in the revised manuscript.

**Changes in manuscript:**

**Results and discussion (Lines 230-231 in the revision):** "There are significant changes of $PM_{2.5}$ between the lockdown and pre-lockdown periods. Moreover, we directly quantify the change in $PM_{2.5}$ during the lockdown."

**Comments:** L229: are lower...

**Response**: Thanks for the comments. The corresponding changes have been made in the revised manuscript.

**Changes in manuscript:**

**Results and discussion (Lines 235-236 in the revision):** "Primary components of $PM_{2.5}$ (EC and POA) are lowered by an average of 37% and 14%, respectively."

**Comments:** L269: this sentence needs a better explanation. You want to demonstrate that HCHO is a good proxy in the model for overall VOC, and it can be observed as well...

**Response**: Thanks for the comments. HCHO is used as a proxy for the total VOCs in accordance with previous studies such as Palmer et al. (2003). Previous studies claimed that HCHO is one of the major contributors to total VOCs reactivity (Zhang et al., 2012; Steiner et al., 2008). Besides, HCHO has a strong correlation with VOC ($R^2$ up to 0.93) and performed well when validated by comparing with satellite-observed data. We made more corresponding explanations in the revised manuscript.

**Changes in manuscript:**

**Results and discussion (Lines 273-276 in the revision):** "We investigated the changes of MDA8 $O_3$ and its major precursors $NO_x$ and HCHO during the lockdown period. HCHO is one of the major contributors to total VOCs reactivity (Zhang et al., 2012; Steiner et al., 2008). It also has a strong correlation with VOC ($R^2$ up to 0.93) (Fig. S4) and performs well when validated by comparing with

satellite-observed data. As a result, HCHO is used as a good proxy in the model for the total VOCs, consistent with previous studies such as Palmer et al. (2003)."

**Comments:** L275-280: this is interesting, but it is not clear if you are suggesting that this is important or not in understanding results.

**Response**: Thanks for the comments. We added more corresponding explanations about the influence of $O_3$ production sensitivity on its concentration during the lockdown. The change of $O_3$ production sensitivity regimes during lockdown played an important role in the change in MDA8 $O_3$ concentration in India. The large reduction of $NO_x$ led to a decrease in MDA8 $O_3$ in most Indian regions that are $NO_x$-limited. While the rise of MDA8 $O_3$ (averaged 5% and up to 21%) was found sporadically in the VOC-limited areas in which more significant decreases of $NO_x$ (compared with VOCs) reduce the $O_3$ consumption ($NO + O_3 = NO_2 + O_2$) and enhance $HO_x$ concentrations result in an increase in $O_3$ levels. Also, the increase in $O_3$ was amplified regionally by the expansion of the VOC-limited zone during the lockdown.

**Changes in manuscript:**

**Results and discussion (Lines 284-292 in the revision):** "Besides, $O_3/NO_y < 6$ indicates that $O_3$ formation is VOC-limited, $O_3/NO_y > 8$ indicates $NO_x$-limited, and intermediate values are transitional. In India, $NO_x$-limited regimes are found in vast areas from both Case 1 and Case 2, which was also reported in previous studies (Mahajan et al., 2015). As a result, the large reduction of $NO_x$ leads to decreased MDA8 $O_3$ in most Indian regions. Compared to Case 1, the VOC-limited area expands mainly in the northwest and south of India from Case 2 during the lockdown. Simultaneously, the rise of MDA8 $O_3$ (averaged 5% and up to 21%) is found sporadically in these VOC-limited areas in which more significant decreases of $NO_x$ (compared with VOCs) reduce the $O_3$ consumption ($NO + O_3 = NO_2 + O_2$) and enhance $HO_x$ concentrations result in an increase in $O_3$ levels. It may also indicate that the increase in $O_3$ is amplified regionally by the expansion of the VOC-limited regimes due to the lockdown."

**Comments:** L302: Please clarify this sentence. You are not comparing pre-lockdown with lockdown, but the emissions scenarios. Correct? Please give a quantification of the importance of meteorology and emission in determining the change.

**Response**: Thanks for the comments. In this sentence, we were comparing pre-lockdown (Case 1) with lockdown (Case 2) to explore the comprehensive effects of meteorology and emissions on the air quality.

The increases of $O_3$ and other key pollutants in some areas show the important role of various meteorological conditions, which has been discussed in Section 3.2 in the manuscript. The comparison of pre-lockdown (Case 1) with lockdown (Case 1) can determine the effects of variation of meteorology on air quality. For example, it can be concluded that the rise of MDA8 $O_3$ in some areas is affected by the increase in temperature (4.1K) (Fig. 4 & Fig. 7 in the manuscript).

However, in a high polluted country like India, the lockdown provides a valuable opportunity to assess air pollutants' changes with significantly reduced anthropogenic emissions in a short time. Our study mainly wants to quantify the change in air quality due to the reduced anthropogenic emissions during the lockdown by comparing Case 1 (without emission reductions) and Case 2 (with emission reductions). Consequently, we didn't add more meteorology impacts analysis in the revision. The specific changes in primary and secondary pollutants can tell the effects of emission reduction on air quality across India, such as a decrease of 15% in MDA8 $O_3$ (Case 2 - Case 1). The corresponding changes have been made in the revised manuscript.

**Changes in manuscript:**

[revised manuscript text omitted]